# Zygotic activation of transposable elements during zebrafish early embryogenesis

Bo Li [1,2,7], Ting Li[3,7], Dingjie Wang[1,2,7], Ying Yang [4,5], Puwen Tan[1], Yunhao Wang [1,2], Yun-Gui Yang [4,5] ✉, Shunji Jia [6] ✉ & Kin Fai Au [1,2] ✉

Although previous studies have shown that transposable elements (TEs) are conservatively activated to play key roles during early embryonic development, the details of zygotic TE activation (ZTA) remain poorly understood. Here, we employ long-read sequencing to precisely identify that only a small subset of TE loci are activated among numerous copies, allowing us to map their hierarchical transcriptional cascades at the single-locus and single-transcript level. Despite the heterogeneity of ZTA across family, subfamily, locus, and transcript levels, our findings reveal that ZTA follows a markedly different pattern from conventional zygotic gene activation (ZGA): ZTA occurs significantly later than ZGA and shows a pronounced bias for nuclear localization of TE transcripts. This study advances our understanding of TE activation by providing a high-resolution view of TE copies and creating a comprehensive catalog of thousands of previously unannotated transcripts and genes that are activated during early zebrafish embryogenesis. Among these genes, we highlight two that are essential for zebrafish development.

Transposable elements (TEs) make up a significant portion of animal genomes (e.g., human, 46%; mouse, 37%; zebrafish, 51%)[1–3], by spreading across the genome in different formats, such as intact sequences with complete transcription units, and fragments inserted into genes or truncated copies within intergenic regions. These confounding formats result in diverse and complicated activities and functions of TEs. In addition to potential transposition activity following autonomous expression (e.g., retrotransposons), TEs are also known to contribute to gene transcription in many aspects, such as TE-derived regulatory elements (e.g., enhancers and promoters), and TE-derived chimeric transcripts with miscellaneous biological functions[4–7]. While dysregulated TE expression has been strongly associated with developmental defects[8–10], there is growing urgency to achieve a more accurate and comprehensive understanding of TEs and their broader biological impact. The community is making progress towards resolving this problem. For instance, recent studies based on single-cell RNA-seq have indicated that cell type-specific activation of different TEs may contribute to tissue-specific developmental processes[11–13]. Recently, the activation of a particular endogenous retrovirus has been implicated as critical for mesoderm development in zebrafish[14].

Previous studies have shown that TE transcription can be also activated as regular genes over early embryonic development in mouse, zebrafish, and a few other organisms[7,10,11,15]. Over this critical developmental course, the degradation of maternal transcripts is sophisticatedly coordinated with the zygotic genome activation, where the activation dynamics of regular genes have been widely studied[16–19]. In contrast, limited details of TE activation and transcription over early embryonic development are known due to (1) the hierarchical sequence similarity within TE families and subfamilies, hindering the single-copy-resolved studies of each locus and transcript[20,21]; and (2) the ambiguity between autonomously expressed

[1]Gilbert S. Omenn Department of Computational Medicine and Bioinformatics, University of Michigan, Ann Arbor, MI, USA. [2]Department of Biomedical Informatics, The Ohio State University, Columbus, OH, USA. [3]School of Life Sciences, Fudan University, Shanghai, China. [4]China National Center for Bioinformation, Beijing, China. [5]Beijing Institute of Genomics, Chinese Academy of Sciences, Beijing, China. [6]Institute of Genetics and Developmental Biology, Chinese Academy of Sciences, Beijing, China. [7]These authors contributed equally: Bo Li, Ting Li, Dingjie Wang. ✉e-mail: ygyang@cncb.ac.cn; jiasj@genetics.ac.cn; kinfai@umich.edu

TE copies and TE-gene chimeric transcripts[10,20]. Therefore, there is a critical knowledge gap of the complicated roles of TE-derived transcripts in transposition activity against genome integrity and gene regulation over embryogenesis[10].

Here we leverage high-quality long reads plus manual annotation to establish a high-resolution landscape of TE activation and transcription at the levels of locus and transcript, over zebrafish early embryonic development. Of note, in this study, TE activation is specifically defined as transcriptional activity, without implying transpositional activity. More importantly, we reveal a previously uncharacterized temporal trajectory and subcellular distribution of zygotic TE activation (ZTA) in zebrafish, where extensive variation exists among TE families, subfamilies, loci, and transcripts with respect to evolutionary age. While ZTA is an integral component of zygotic genome activation (ZGA) and may rely on zygotic gene activation, unique features are identified compared with regular genes in terms of their transcription dynamics and transcriptional regulations during zebrafish early development. Additionally, this unambiguous transcriptome catalog lists out thousands of previously unannotated transcripts and genes, among which two have been experimentally verified to be essential for zebrafish early embryogenesis. This comprehensive approach provides insights into the role of TEs in early developmental processes.

## Results

### Only a tiny subset of TE-alone loci are actively transcribed over zebrafish early embryogenesis

A total of 25,788 full-length TE-derived transcripts are identified from 11 featured stages (from fertilization to the shield stage), including 706 autonomously expressed TE transcripts (referred to as "TE-alone") and 25,082 chimeric transcripts of TE and gene sequences (referred to as "TE-gene") (Fig. 1a, b). The TE-alone transcripts undergo stringent manual curation that merges highly identical transcripts from the same TE loci, corrects TE annotation and transcript structures, and removes incorrectly annotated SINEs (short interspersed nuclear elements) (Fig. 1a, Supplementary Fig. 1 and Supplementary Note 1). The manual annotation confirms that 706 active TE-alone transcripts are expressed at 550 TE-alone loci, from 30 families and 210 subfamilies (Fig. 1c, d and Supplementary Fig. 2). Despite most TE-alone transcripts deriving from single TE loci, 48 chimeric TE-alone transcripts are likely generated from multiple adjacent loci by readthrough transcription and nested TE insertion (Fig. 1e, Supplementary Fig. 3, Supplementary Table 1 and Supplementary Note 2).

Compared to the short read-based characterization of expressed TE families/subfamilies containing numerous loci in zebrafish and mouse[11,22], manual annotation with the unambiguous long-read RNA-seq alignment constructs an accurate and high-resolution catalog of active TE-alone loci and transcripts over zebrafish early embryogenesis (Supplementary Fig. 4 and Supplementary Note 3). Therefore, the expression dynamics from a small subset of TE loci would not be mixed or overwhelmed by the tremendous number of inactive copies across the whole genome. The high-resolution catalog reveals the distinct splicing patterns of DNA transposons, LINEs (long interspersed nuclear elements), and LTRs (long terminal repeats) (Fig. 1f and Supplementary Table 2). A total of 95.13% (215/226) of DNA transposon transcripts contain multiple exons, compared to 17.05% (15/88) of LINE transcripts. In between, 166 and 224 LTR transcripts are expressed with/without exon splicing, respectively (Fig. 1f and Supplementary Fig. 5a, b). Some introns in active LINE and LTR loci are acquired by insertion of other TEs (Supplementary Fig. 5c and Supplementary Note 4). A total of 85.39% of splicing sites from the active TE-alone transcripts contain canonical splicing signal GU-AG, and this bias is more significant in DNA transposon transcripts than LTR ones and LINE ones (Fig. 1f and Supplementary Fig. 5a). A total of 31.50% of active DNA transposon loci express multiple transcript isoforms, significantly

higher than 8.86% for LINEs and 11.11% for LTRs (Fig. 1f and Supplementary Fig. 5a).

With the full-length sequences, we further investigate whether the expressed TE-alone transcripts encode essential proteins to fulfill the transposition process[23]. Most LINE transcripts may maintain autonomous transposition potential, as only 13 (14.77%) lose one or two core domains (Fig. 1g). In contrast, 178 (78.76%) DNA transposon transcripts and 366 (93.85%) LTR transcripts lose core domains. Particularly, all 170 active ERV transcripts, accounting for 43.59% of total LTR transcripts, lack one to six core domains, suggesting completely or partially deficient autonomous transposition function (Fig. 1g and Supplementary Table 3).

### ZTA shows a unique activation pattern during zebrafish early embryogenesis

Previous studies reported that TE is also activated as zygotic gene activation in mammals and zebrafish[7,11], yet the precise activation timing, trajectory and variation at the levels of locus and transcript are unknown. Among 706 active TE-alone transcripts, only 11 (including 9 DNA transposons), are maternally inherited (Supplementary Fig. 6a), and 695 zygotic ones undergo considerably later activation than the ZGA waves in zebrafish (Fig. 2a). The ZTA starts at the 1k-cell stage with 28 TE-alone transcripts (mean TPM = 2.07), and has a remarkable increase at the oblong stage with a greater number of transcripts (108) and higher abundance (mean TPM = 5.52) (Fig. 2a, Supplementary Fig. 6b and Supplementary Note 5), which is supported by the stage-by-stage expression correlation analysis (Fig. 2b and Supplementary Fig. 6c) and whole mount in situ hybridization (WISH) (Fig. 2c and Supplementary Fig. 6d). By contrast, previous studies have indicated that the minor wave of ZGA typically initiates between the 64-cell and 512-cell stages, while the major wave predominantly begins at the 1k-cell stage[24–26]. Moreover, the expression levels of TE-alone transcripts differ significantly from those of TE-gene and regular gene transcripts during the developmental window spanning the 1k-cell to shield stages (Fig. 2a). In particular, before the oblong stage, TE-alone transcripts exhibit significantly lower expression levels compared to TE-gene and regular gene transcripts, but after the oblong stage, their expression levels increase sharply, surpassing those of both TE-gene and regular gene transcripts (Fig. 2a).

The difference between the existing knowledge and our findings likely arises from the limitation of the conventional short read-based analyses, where TE-gene transcripts and TE-alone ones are not distinguished unambiguously: using short reads alone, we could only show a similar activation curve between TEs and genes (Supplementary Fig. 6e). In fact, 71.21% of TE-derived transcripts contain <20% TE proportion and their activation pattern is similar to ZGA. When the short read-based analysis pooled TE-derived transcripts, the overall pattern was demonstrated by this subset of TE-derived transcripts and thus was similar to ZGA.

We find that ZTA in mouse shares a similar delayed onset by analyzing two publicly available data[27,28]: while ZGA can be detectable as early as 4 h post fertilization (hpf), the first TE-alone transcripts (i.e., from the subfamilies L1MdTf_II, L1MdTf_III and L1MdA_II) are activated at 6 hpf, followed by MERVL and MERVL-3A at 8 hpf and MT2_Mm at 10 hpf ("TE activation during early mouse embryonic development" subsection of the "Methods" and Supplementary Fig. 7). From 6 to 12 hpf, the overall TE-alone transcriptome remains at a certain level without remarkable change (Supplementary Fig. 7) while 1777 genes are upregulated[27]. Because the abovementioned mouse TE quantification is based on a short-read RNA-seq dataset (from 0 to 12 hpf)[27], we could only investigate mouse ZTA at the subfamily level until 12 hpf. A more comprehensive long-read RNA-seq data collection at fine time points beyond 12 hpf until the 2-cell stage is required to further reveal the details of the species specificity.

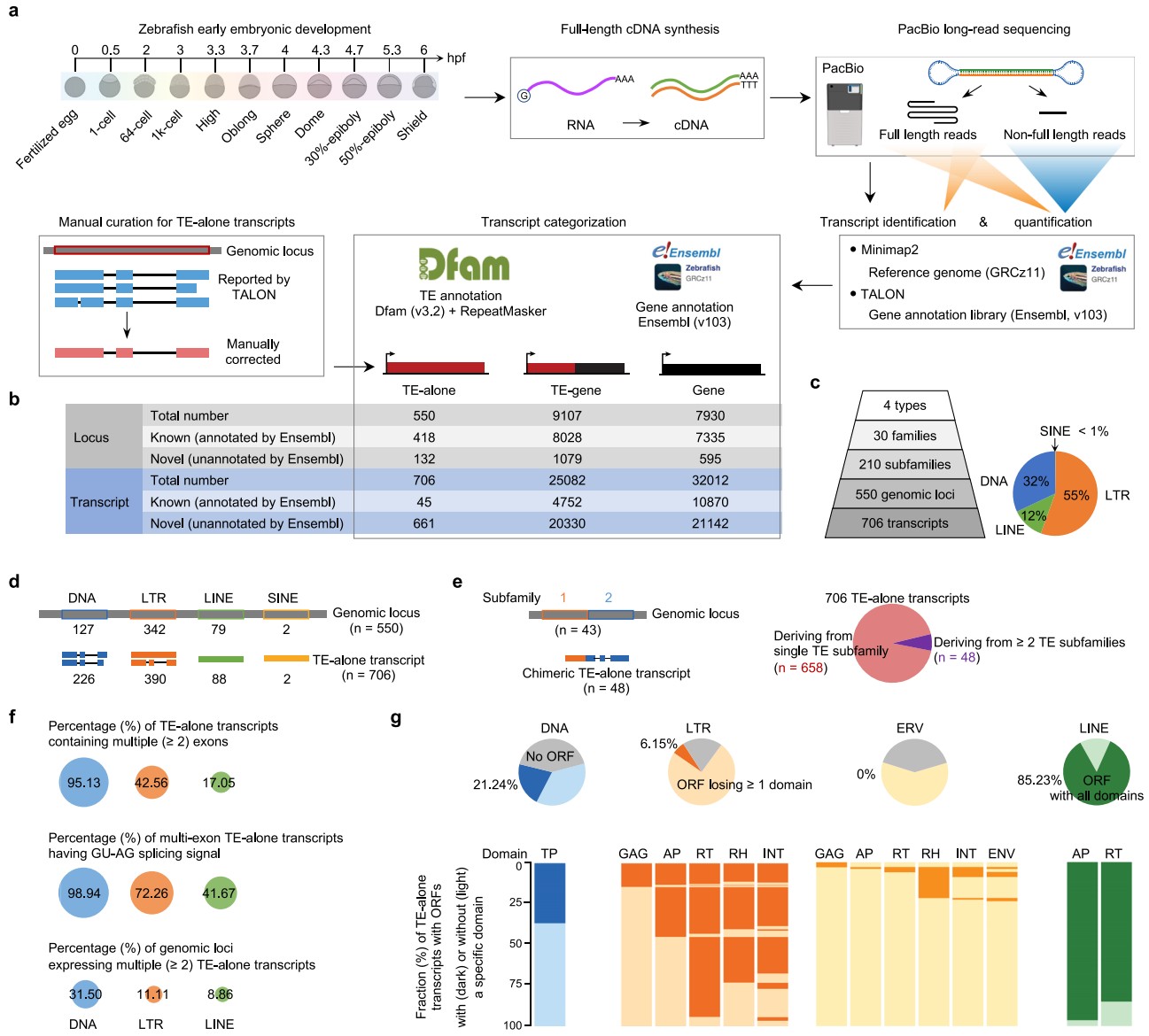

**Fig. 1 | Characterization of TE-derived transcripts. a** Experimental and bioinformatics analysis pipeline to identify and classify TE-derived transcripts over zebrafish early embryonic development. Total RNA is extracted from 11 developmental stages (hpf hours post fertilization) and reversely transcribed into cDNA for PacBio RNA sequencing. Full-length (FL) circular consensus sequence (CCS) reads are used for transcript identification with TALON and both FL and Non-FL CCS reads are used for quantification. Gene and TE annotations are used for classifying all transcripts into TE-alone, TE-gene and gene subgroups. TE-alone, autonomously transcribed TE transcript; TE-gene, chimeric transcript of TE and gene; gene, TE-sequence-free transcripts. Manual curation is used for further polishing the TE-alone transcript annotation. **b** Number of identified loci and transcripts among three different types. **c, d** Categorization of 706 TE-alone transcripts based on TE type/family/subfamily/loci. **e** Characterization of chimeric TE-alone transcripts.

**f** Transcriptional patterns among three TE types. Comparisons are conducted with differences in exon number, canonical splicing signal (GU-AG) and alternative splicing. **g** Number of TE-alone transcripts having core conserved domains. Pie charts show the classification of TE-alone transcripts due to the presence and absence of core domains. Gray color, no open reading frames (ORFs); light colors, having ORF but losing domains; dark colors, having all conserved domains. The percentages (%) indicate the proportions of TE-alone transcripts maintaining all conserved domains. Heatmap showing the details of domain presence and absence. Each column represents one domain, and each row represents one transcript (with complete ORF), with light colors indicating domain loss. TP transposase, APE apurinic endonuclease, RT reverse transcriptase, GAG capsid protein, AP aspartic proteinase, RH RNase H, INT integrase, ENV envelope protein[23].

Previous studies have underscored the pivotal role of the minor wave of ZGA in establishing the major wave of ZGA in both zebrafish and mouse[29–31]. Therefore, the delayed onset of ZTA suggests that the preceding ZGA may also provide essential factors for the subsequent ZTA. We revisit a previous study and find transcriptomics evidence to support this possibility from several zebrafish mutants and embryos exposed to transcription/translation inhibitors[29]: ZTA can be abolished through α-Amanitin injection (Fig. 2d) or repressed in triple loss-of-function of Nanog, Pou5f1, and SoxB1 (Fig. 2e and Supplementary Fig. 8), but ZTA can also undergo a global reduction when there is a

translational blockade of the zygotic genes in the minor ZGA wave (Fig. 2f). Furthermore, we look into two transcription factors (TFs) in zebrafish, *mxtx1/2*, which are predicted to bind to the promoter regions (5′ LTRs) of eight ERV subfamilies (Supplementary Fig. 9a). They are orthologous to human *DUX4* and mouse *Dux* that have been well documented for their roles in activating HERVL and MERVL retrotransposons, respectively[32,33]. *Dux* is activated at 6 hpf in mouse embryos, while Dux-responsive genes and TEs are expressed later (e.g., MERVL expressed at 8 hpf) (Supplementary Fig. 7)[27]. In zebrafish, *mxtx1/2* are strictly zygotic genes activated during the 64-cell to 1k-cell

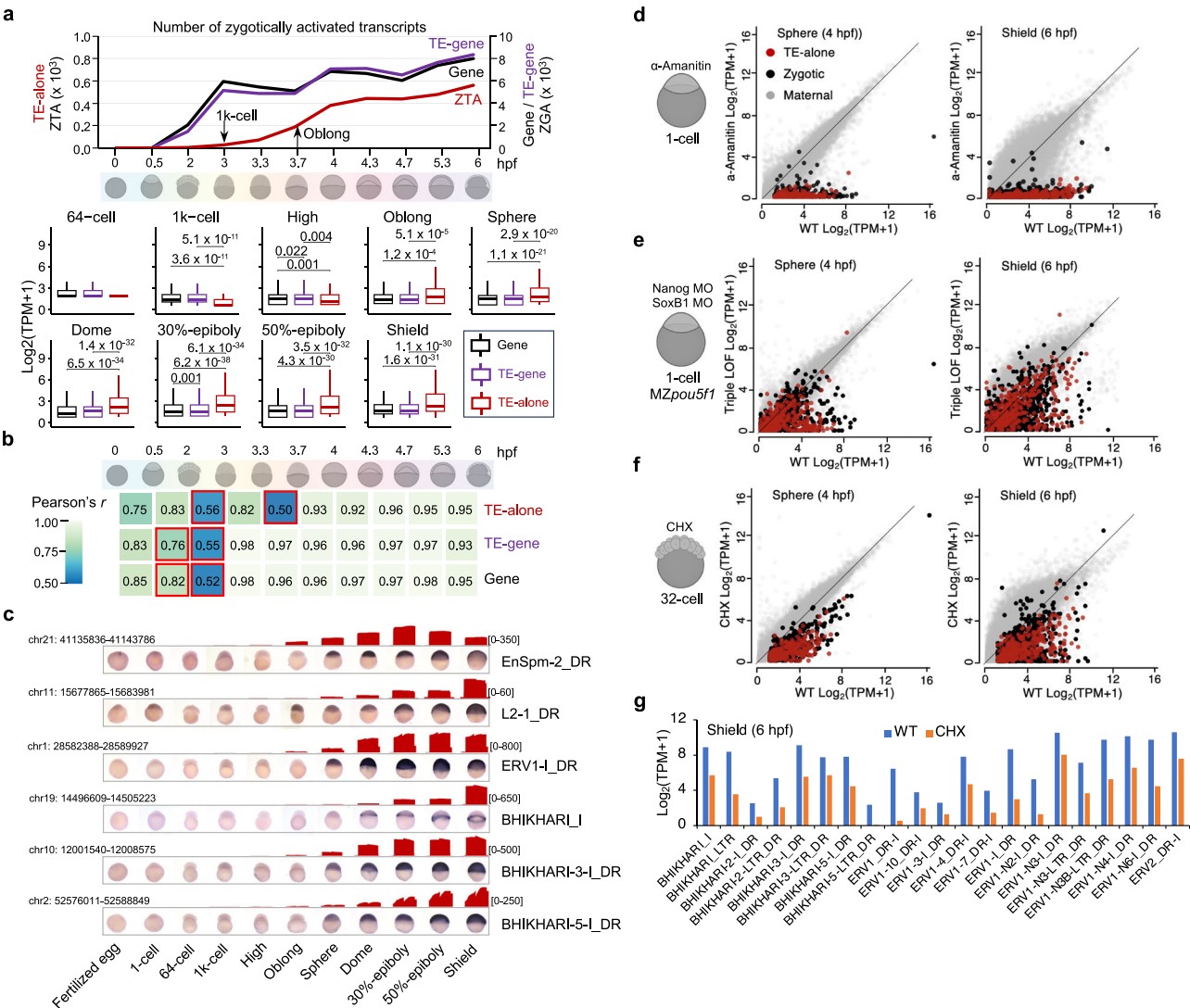

**Fig. 2 | Characterization of ZTA during zebrafish early embryogenesis.**
**a** Number and abundance of zygotically activated TE-alone, TE-gene and gene transcripts across the 11 developmental stages. Two zygotic TE activation (ZTA) stages (1k-cell and oblong) are marked by arrows. Boxplots show expression distribution of each type of transcripts from the 64-cell to shield stages. Wilcoxon rank sum tests, two-sided are used to test the significant differences among gene, TE-gene and TE-alone. The total number of transcripts at each stage is 3563, 11,132, 10,415, 10,187, 14,352, 14,275, 13,050, 15,539, 16,936 in the corresponding boxplots.
**b** Heatmap showing the expression similarity (Pearson's correlation) between adjacent developmental stages for TE-alone, TE-gene, and gene transcripts, respectively, with red boxes highlighting the low similarity. **c** WISH experiments

showing the activation stages with the representative TE subfamilies. Abundances estimated by long reads are drawn for each subfamily with the bar plot.
**d** Widespread TE-alone and gene expression loss after α-Amanitin treatment at the sphere and shield stages. WT wide type, Zygotic strictly zygotic genes, Maternal maternal genes. **e** Widespread TE-alone and gene expression loss in the triple LOF (loss-of-function: maternal zygotic MZ*pou5f1* mutant with translation-blocking morpholinos for Nanog and SoxB1) at the sphere and shield stages. MO morpholino, MZ maternal zygotic. **f** Widespread TE-alone and gene expression loss at the sphere and shield stages following CHX (cycloheximide) treatment to block translation of zygotic genes. **g** Downregulated zygotic activation in zebrafish ERV subfamilies at the shield stage following CHX treatments.

stages, preceding the defined ZTA period (i.e., the 1k-cell to oblong stages) (Supplementary Fig. 9b). Consequently, after blocking the translation of *mxtx1/2* (Supplementary Fig. 9c) and other early zygotic genes, all 21 ERV subfamilies are downregulated as compared to the wild type at the sphere (Supplementary Fig. 9d) and shield stages (Fig. 2g). Interestingly, re-analysis of the publicly available ChIP-seq data for zebrafish Nanog and Mxtx2[34] suggests that Nanog may function as a general TF for various types of TEs, while Mxtx2 may specifically regulate ERVs (Supplementary Fig. 9e, f).

**Programmed ZTA shows complex expression trajectories during the maternal-to-zygotic transition**
From the maternal carryover in fertilized egg to the zygotic transcripts in the shield stage, the total abundance of TE-alone transcripts

increases by 318 folds (Fig. 3a), and there is a dramatic composition shift of TE-alone transcriptome components: DNA transposons account for the majority of the maternally inherited transcripts until ZTA starting at the 1k-cell stage, while ZTA of LINEs and LTRs, especially ERVs, rapidly dominate TE-alone expression (Fig. 3a and Supplementary Fig. 10a). However, considerable variations on expression pattern, including the abundance, activation time and peak stages, exist extensively among their families/subfamilies (Fig. 3b, Supplementary Fig. 10a, b and Supplementary Note 6). For example, the WISH experiment validates the activation of two ERV1 subfamilies BHIKHARI-3-I_DR and BHIKHARI-5-I_DR at the sphere stage while the other subfamily BHIKHARI_I at the dome stage (Fig. 2c).

In a higher-resolution view, these variations become more remarkable and extensive at both levels of locus and transcript within

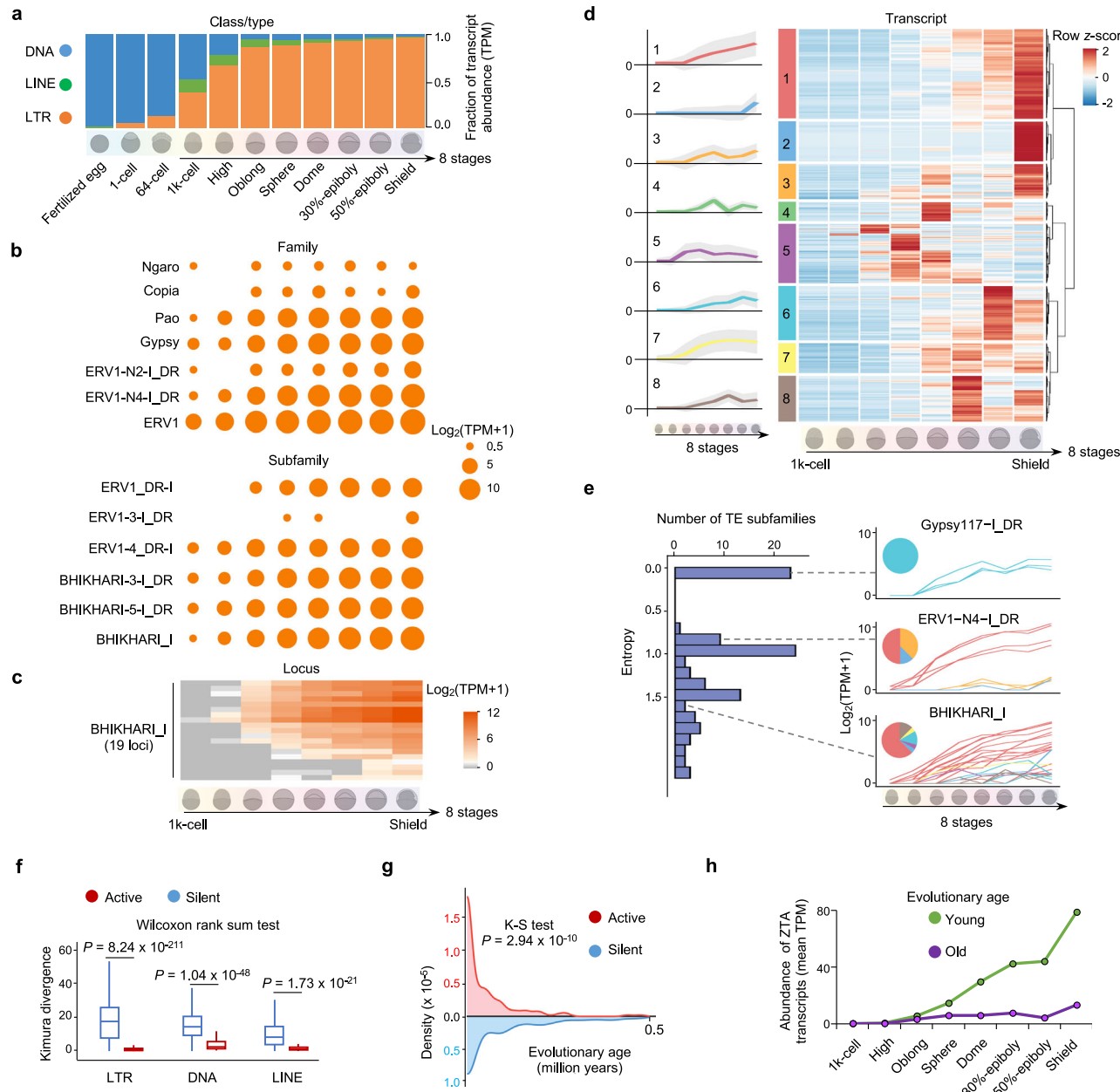

**Fig. 3 | Expression dynamics of TE-alone transcripts during zebrafish early embryonic development. a** Fraction of transcript abundance of DNA transposons, LINEs and LTR over 11 developmental stages. **b** Dynamic activation and transcription pattern of seven representative LTR families (top) and six ERV1 subfamilies (bottom). **c** Heatmap showing the differential activation and expression of 19 loci from BHIKHARI_I subfamily. **d** Heatmap (right) showing the expression of 695 ZTA TE-alone transcripts, categorized into eight groups with hierarchical clustering. Line plots (left) showing the overall/merged expression trend (with the colored line as mean value and the gray band as standard deviation value) across all transcripts in each group. **e** Distribution of entropy (which is used to measure the complexity of expression pattern of TE-alone transcripts/loci from the same subfamily) of each TE subfamily based on the number of patterns. Pie charts representing the transcript constitution of three representative TE subfamilies. The color of the pie chart is consistent with the expression clusters shown in (**d**). Line plots showing the expression pattern of transcripts from corresponding subfamilies during early embryonic development. **f** Comparison of Kimura divergence between active and silent TE copies among different TE types. Wilcoxon rank sum test, two-sided was applied. The number of silent and active copies is 29,235 and 342 for LTR, 172,202 and 127 for DNA transposon, and 35,795 and 79 for LINE. **g** Density plot for comparing the LTR insertion years between expressed and not expressed intact copies which are *do novo* identified. K-S Kolmogorov-Smirnov test, two-sided. Insertion years larger than 0.5 million years are not shown. **h** Line plot to show expression pattern of evolutionarily young and old TE-alone loci.

subfamilies (Fig. 3c, d and Supplementary Fig. 11a). Among 97 and 103 TE subfamilies with multiple active TE-alone loci and transcripts, respectively, 79 and 86 subfamilies have ≥2 activation time points for different loci and transcripts (Supplementary Fig. 11b). The activation trajectories of all 695 zygotic TE-alone transcripts are categorized into 8 types (Fig. 3d). The heterogeneity of ZTA trajectories within each subfamily, which is measured by entropy ("Expression trajectory

analysis on the locus/transcript level" subsection of the "Methods"), varies greatly from being homogeneous to containing 7 of 8 types with comparable proportions (Fig. 3e). 72 and 80 subfamilies with multiple active TE-alone loci and transcripts, respectively, span ≥2 types of ZTA trajectories (Supplementary Fig. 11c). This intra-subfamily hetero-geneity and the inter-subfamily variation exist extensively in TE-alone expression of DNA transposons, LTRs and LINEs (Supplementary

Fig. 12a). For example, within the BHIKHARI_I subfamily, 24 active TE-alone transcripts belong to 6 trajectory types, including some activated as early as at the 1k-cell and high stages while some others as late as at the 30%-epiboly stage; some continuously increasing and some others saturating or even decreasing prior to the shield stage (Fig. 3e and Supplementary Fig. 12a). Although WISH could reveal the subfamily-level difference between BHIKHARI_I versus BHIKHARI-3-I_DR and BHIKHARI-5-I_DR (Fig. 2c), WISH and the other conventional approaches rarely disclose the detailed complexity among the activated transcripts within each subfamily (Fig. 3e). However, 19 active loci of these 24 TE-alone BHIKHARI_I transcripts contain nearly identical regulatory sequences that were reported previously[35], including the conserved CCAAT and TATA boxes at the promoter regions (Supplementary Fig. 12b). The intra-subfamily heterogeneity of their ZTA trajectories versus the high similarity of the annotated regulatory elements highlights the different epigenetics layers of regulation and the varying genomic contexts in the locus-/transcript-specific manner over zebrafish embryogenesis.

### Evolutionary age influences ZTA

The embryogenesis activation pattern of TE-alone loci varies with respect to their evolutionary ages. Kimura divergences (KDs) of the active LTR loci are significantly lower, indicating younger evolutionary ages, than the silent copies (Fig. 3f). This significant difference extends to the active copies versus the silent ones of the intact LTRs (Supplementary Fig. 13a), which is supported by the orthogonal measurement of evolutionary age, i.e., the estimated insertion years (Fig. 3g and "Evolutionary analysis on TE-alone loci" of the "Methods"). Similarly, the active DNA transposon and LINE loci are also significantly younger than the silent ones (Fig. 3f).

Despite that there is no linear correlation between KD and the abundance of the active TE-alone loci (Supplementary Fig. 13b), there exist critical differences in the activation trajectories between the relatively "young" and "old" subsets that are defined by comparative genomics (Supplementary Fig. 14a, b and Supplementary Note 7). The evolutionarily young TE-alone loci show higher and increasing abundance at the oblong stage, while the abundance of the old subset remains at a certain level after activation (Fig. 3h and Supplementary Fig. 14c, d). The majority (50.60%) of the young subset are activated at the oblong and sphere stages, while the activation time of the old ones spreads more evenly across eight stages (Supplementary Fig. 14e). In addition, two subsets are distributed differently in eight expression clusters (Supplementary Fig. 14f).

### Imbalanced subcellular localization of TE-derived transcripts

RNA subcellular localization is closely associated with diverse biological functions[36], which can be exemplified by the comparison between long intergenic non-coding RNAs and protein-coding genes (Supplementary Fig. 15a). In the context of three stages over ZTA (1k-cell, dome and shield), TE-alone transcripts have the highest relative mRNA abundance between the nucleus and cytoplasm (N/C proportion) at the subfamily level, followed by TE-gene transcripts, and gene transcripts have the lowest ratio (Fig. 4a, b and "Subcellular localization analysis with nuclear and cytosolic RNA-seq data" subsection of the "Methods"). TE-gene transcripts display a mixed pattern, with a relative N/C proportion similar to that of genes at the 1k-cell stage but shifting towards the TE-alone transcripts' ratio at the shield stage (Fig. 4b). Moreover, the relative N/C proportion of a transcript is positively correlated with its TE fraction (Fig. 4c), highlighting a possibility of TE sequence-mediated regulation of transcript subcellular localization[37,38].

At the resolution of TE subfamily, significant variations in the relative N/C proportion are observed across different developmental stages. Despite these variations, DNA transposons consistently have a lower relative N/C proportion compared to LTRs and LINEs (Fig. 4d),

suggesting a milder nuclear bias for DNA transposons. Additionally, LTRs show significantly higher relative N/C proportion than LINEs at the dome stage, but this pattern reverses at the shield stage, indicating dynamic subcellular localization for different TE subfamilies during development.

To illustrate these observations, we applied fluorescence in situ hybridization (FISH) to investigate the dynamic subcellular localization of several TE subfamilies. Our analysis reveals that the DNA transposon subfamily hAT-N76_DR predominantly localizes in the cytoplasm, while the LINE subfamily L2-1_DR is localized in the nucleus, consistently from the shield stage to the 6-somite stages (Fig. 4e, f and Supplementary Fig. 15b, c). These findings align with a previous study in mouse embryonic stem cells, which demonstrated a nuclear bias of LINE1 transcripts, suggesting potential regulatory roles in early development[39]. This parallel implies a similar function for LINE transcripts in zebrafish embryogenesis. The transposition mechanisms likely play a key role in influencing the subcellular localization of TEs. For instance, LINEs replicate via target-primed reverse transcription, requiring their mRNA template to be transported back into the nucleus. This process may likely contribute to the observed N/C proportion[40]. In addition, the LTR subfamily BHIKHARI_I displays a more dynamic subcellular distribution across developmental stages. It shows a nuclear bias at the shield stage but shifts to clear cytoplasmic localization at the bud and 6-somite stages (Fig. 4g and Supplementary Fig. 15d). This dynamic localization pattern aligns with above speculations (Fig. 4d), highlighting a more flexible subcellular positioning for LTRs during development.

### Epigenetic dynamics over the full course of ZTA

Several types of ZTA trajectories (Type 4 to 8 in Fig. 3d) are saturated or even decrease by the shield stage, which may reflect a control of their transposition potential against genome integrity. Indeed, the whole TE-alone transcriptome landscape over zebrafish embryogenesis includes not only activation but also a repression process at the gastrula and segmentation stages (Fig. 5a). Although the accurate timing of repression may be heterogeneous as their activation timing, the full course is coordinated with epigenetics changes at multiple layers (Fig. 5b).

Previous studies have revealed that the establishment of an open/accessible chromatin status at gene promoters typically marks future transcriptional activation in many species[24]. To investigate the epigenetic regulation of TEs, we analyzed the publicly available datasets of a few epigenetic marks[41–45]. The chromatin accessibility around the TSSs of 550 active TE-alone loci increases since the high and oblong stages and decreases at the shield and 80%-epiboly stages (Fig. 5c), which are immediately followed by the increase and decrease of their overall abundance, respectively (Fig. 5a). In contrast, the chromatin openness of regular genes increases continuously from the 64-cell to the 80%-epiboly stage (Supplementary Fig. 16a and Supplementary Note 8).

As previously reported[46], the active mark H3K4me3 exists in sperm yet is removed in the early embryo for both genes and TEs (Supplementary Fig. 16b and Fig. 5d, respectively). Compared to regular genes, which show elevated levels of H3K4me3 at the 128-cell stage, TE-alone loci have a subtle increase at the 1k-cell stage and a significant rise at the dome stage (Fig. 5d, e and Supplementary Fig. 16b, c), which is consistent with the delayed activation of TEs versus genes. Another active mark H3K27ac also demonstrates a similar regulatory pattern (Fig. 5f and Supplementary Fig. 16d). The repressive mark H3K9me3 is deposited across TE-alone loci at the sphere and shield stages (Fig. 5b, g), prior to their abundance decrease. In addition, as the other epigenetic and post-transcriptional silencing regulators, a rise of PIWI-interacting RNAs (piRNAs) at 76.36% (420/550) of TE-alone loci occurs from the oblong to 50%-epiboly stages (Fig. 5h, Supplementary Figs. 16e and 17 and Supplementary Note 9), suggesting that TE-alone transcript degradation might be mediated by

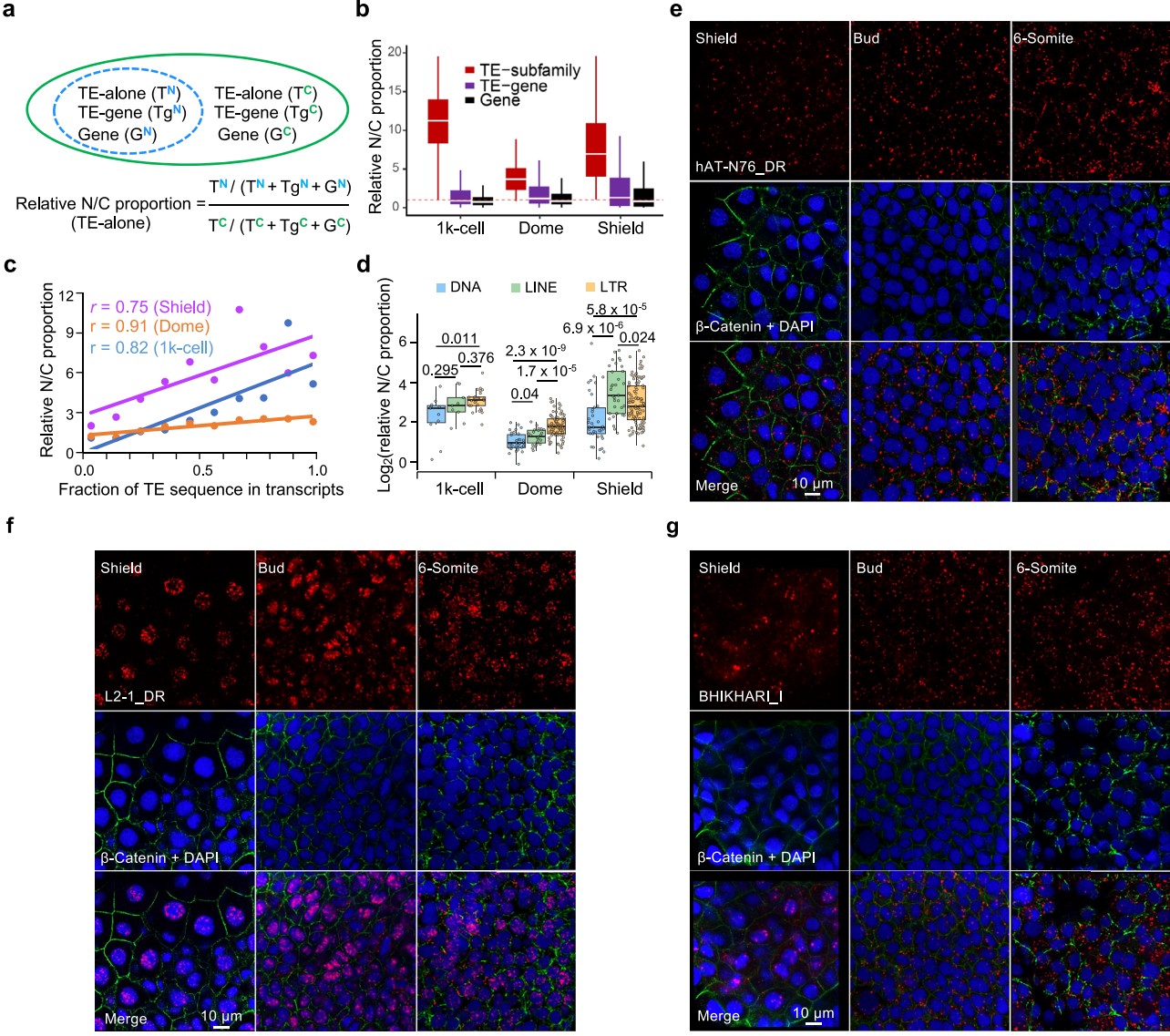

**Fig. 4 | Analysis of subcellular localization of TE-alone loci. a** A chart to illustrate how to calculate the relative N/C proportion for TE-alone, TE-gene and gene transcripts. $T^{N/C}$, $Tg^{N/C}$ and $G^{N/C}$ are total TPMs for TE-alone, TE-gene and gene transcripts in the nucleus and cytoplasm, respectively. **b** Comparison of relative N/C proportion among TE-alone, TE-gene and gene at indicated stages. The red dash line indicates the relative N/C proportion as 1. The outliers and data points with the relative N/C proportion >20 are not shown. The total number of transcripts at each stage is 9302, 10,455, 10,207 (Gene) and 9503, 11,437 and 11,037 (TE-gene). The number of TE subfamilies is 45, 147 and 180. **c** The linear relationship between relative N/C proportion and fractions of TE sequences in TE-gene chimeric transcripts at indicated stages. **d** Comparison of relative N/C proportion among TE subfamilies at indicated stages. Wilcoxon rank sum test, two-sided is applied to statistical significance. The number of DNA transposon, LINE and LTR subfamilies is 11, 9 and 25; 32, 24 and 91; and 38, 34 and 108 for the 1k-cell, dome and shield stages. **e–g** FISH experiments showing the subcellular distribution of transcripts from the three selected TE subfamilies at indicated stages ($n = 3$ independent experiments; 30 embryos were used in total). Green, β-Catenin staining for cell membrane; Blue, DAPI staining for nucleus; Red, specific probes against indicated TE-alone transcripts. Scale bar = 10 μm.

PIWI-piRNA complexes[47]. Moreover, several key factors of the piRNA-medicated regulation, such as *piwil1* and *piwil2* genes, are constantly expressed during the activation process and fade away during the repression process (Supplementary Fig. 16f). Both repressive markers H3K9me3 and small RNA profiles show no obvious control at gene loci, which is distinct from TE-alone loci (Supplementary Fig. 16g, h). In sum, the activation/repression of TE-alone loci and transcripts could be a complex process in multiple pre-/post-transcriptional regulatory layers.

### Characterization of TE-gene chimeric transcripts

In addition to TE-alone transcript, TE-gene chimeric transcript is another critical format that TE sequences contribute to the embryogenesis transcriptome complexity. Many TE-gene chimeric transcripts have been shown diverse and important functions under various biomedical contexts, such as developmental biology and stem cells[6,7,48,49], yet their repetitive fragments lead to the challenge of being discovered or studied precisely by the conventional approaches (Supplementary Fig. 18a). Over the zebrafish early embryogenesis, we identify 25,082 TE-gene transcripts, including 4752 annotated and 20,330 novel transcripts that are supported by at least two high-quality reads (for 24,940 transcripts) or detected in at least two developmental stages (for 23,551 transcripts) (Fig. 1b and Supplementary Fig. 18a, b). Of note, DNA transposon sequences are found in the majority (21,187) of these chimeric transcripts (Supplementary Fig. 18c).

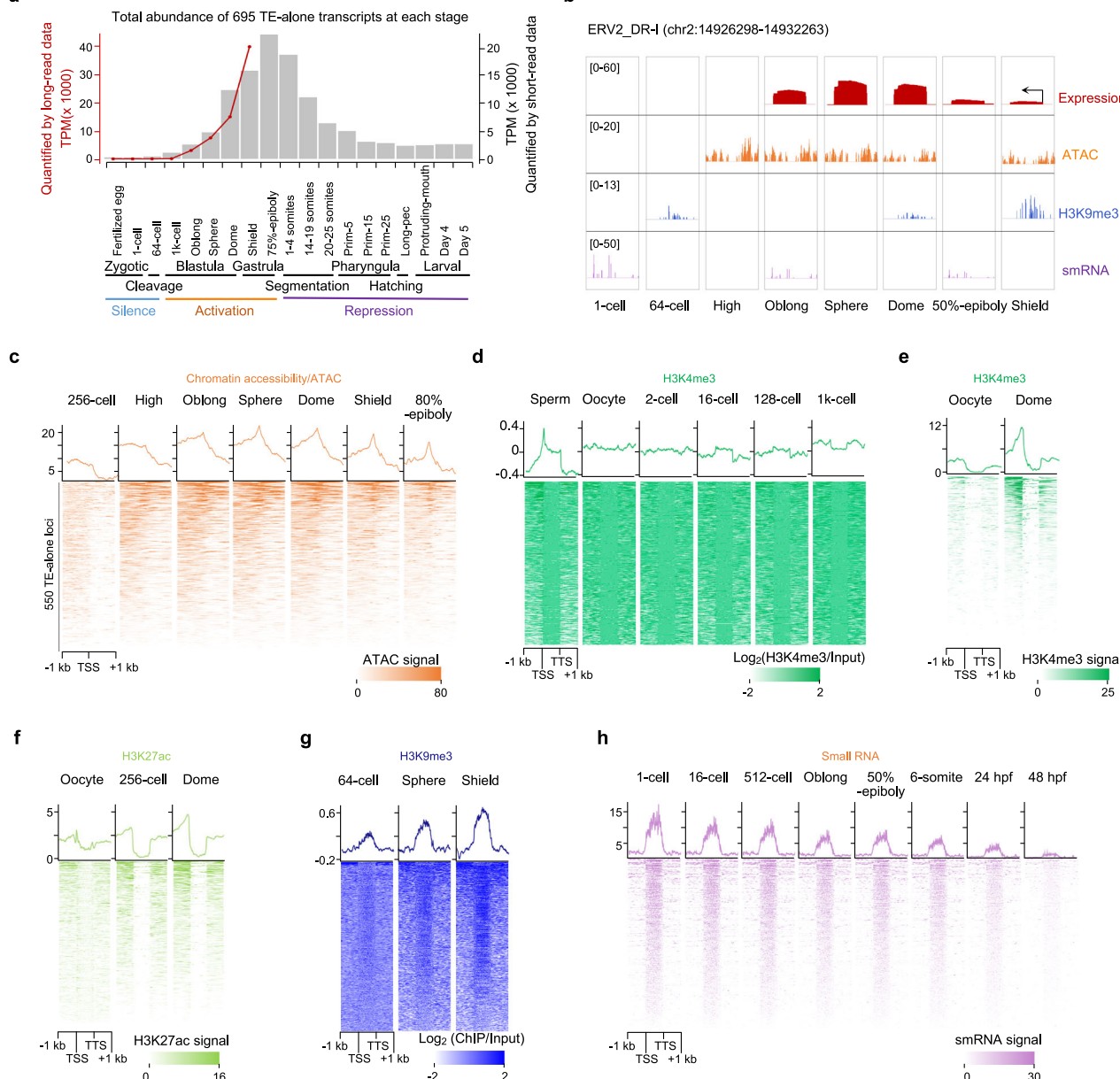

**Fig. 5 | Epigenetic regulation of TE transcriptional activity during zebrafish embryonic development. a** Activation and repression dynamics of TE-alone transcripts measured by long read (line plot) and short read (bar plot) data. Three phases of TE activity during zebrafish early development are defined: silence (zygotic and cleavage), activation (blastula and gastrula) and repression (from segmentation to larval). **b** An example showing the epigenetic profiles over a dynamically activated ERV2_DR-I locus. **c** Dynamic chromatin state changes (ATAC-seq signal) surrounding transcription start sites (TSS) of TE-alone loci. **d, e** ChIP-seq signal of an active mark, H3K4me3 along the TE-alone loci. **f** ChIP-seq signal of an active mark, H3K27ac along the TE-alone loci. **g** ChIP-seq signal of the repressive mark H3K9me3 along the TE-alone loci. **h** Expression dynamics of small RNAs that are aligned onto TE-alone loci. Of them, 56.06–82.84% are piRNAs as defined in "Epigenetic regulation on activation and repression of TE-alone loc" subsection of the "Methods". TTS transcription termination sites.

## Loci identified with essential functions for zebrafish early embryogenesis

Our comprehensive transcriptome analysis of zebrafish early development not only provides a high-resolution profiling of TE activation but also identifies 1674 novel gene loci (including TE-genes and regular genes), that were previously unannotated in the Ensembl reference annotation library[50], which may play critical unknown functions in zebrafish development. Among these, 1260 loci have protein-coding potential, and 698 exhibit known functions and/or conserved domains (Supplementary Fig. 18d, e). To further understand the biological functions of these genes, co-expression network analysis is applied to cluster them with other functionally annotated genes, and two co-expression modules (M1 and M2) were selected for further investigation (Fig. 6a).

Within M1, we identified a locus, *zeat1* (Zebrafish Embryogenesis Associated Transcript 1), serves as a hub gene. *zeat1* is highly abundant at the fertilized egg and 1-cell stages, but degrades since the 64-cell stage (Fig. 6a, b and Supplementary Note 10). M1 genes are maternally inherited, and they are functionally enriched in protein transport, cell cycle, and transcription regulation (Supplementary Fig. 19a), including the important embryonic development genes, e.g., *phf8* essential for cell cycle[51], and *e2f4*, a key transcription activator[52]. A Tc1N1_DR transposon is inserted at the 3′ UTR of *zeat1* without interrupting the coding sequence and four isoforms are identified over early embryogenesis (Fig. 6a). *bmb*, which is essential for mediating nuclear envelope fusion[53], is the only homolog of *zeat1* despite their long-term divergence (Supplementary Fig. 19b). The zygotic *zeat1* (*zeat1−/−*)

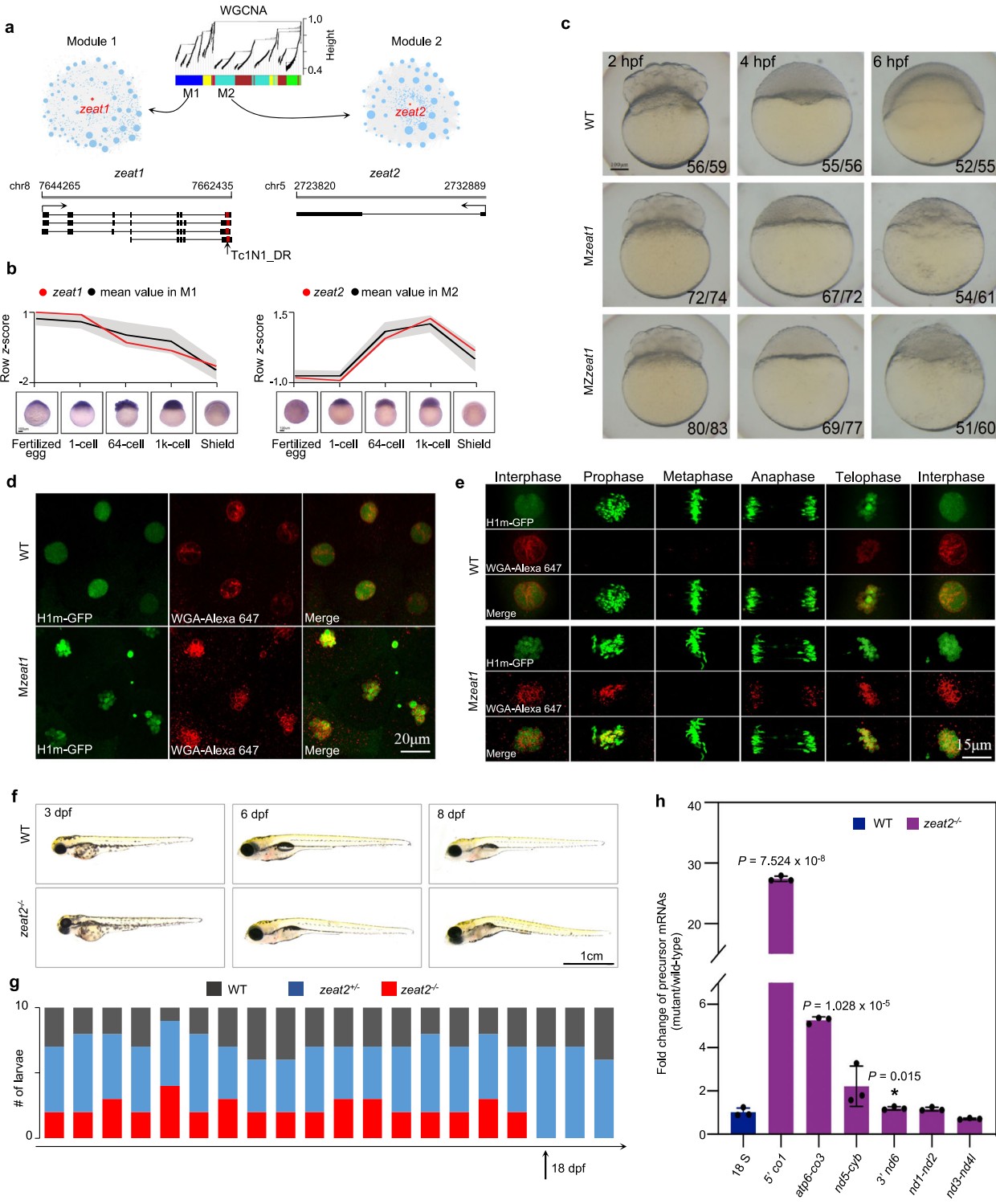

CRISPR-Cas9 knockout mutants (Supplementary Fig. 19c, d) show normal morphology during embryogenesis, while both the maternal *zeat1* (M*zeat1*) and maternal-zygotic *zeat1* (MZ*zeat1*) mutant embryos fail to undergo cell movements associated with epiboly and gastrulation, and arrest shortly after the mid-blastula transition (Fig. 6c). Therefore, *zeat1* is a strictly maternal-effect gene and essential for early embryogenesis (Supplementary Note 10).

To elucidate the function of the *zeat1* gene, we performed microinjections of H1m-GFP protein (a zebrafish H1 type linker histone that labels the chromatin) and wheat germ agglutinin (WGA) Alexa Fluor 647 conjugate (WGA-Alexa 647) into wild-type (WT) and M*zeat1*

mutant embryos at the one-cell stage. These injections labeled the nuclei and nuclear envelope, respectively. Time-lapse confocal microscopy revealed that M*zeat1* mutant embryos showed severely abnormal nuclear morphology that was characterized by fragmented chromatin enclosed by the nuclear envelope (Fig. 6d).

Further investigation of the entire cell cycle suggested that karyomere fusion was disrupted in M*zeat1* mutants, resulting in multi-micronucleated cells during the transition from telophase to interphase (Fig. 6e and Supplementary Movies 1, 2). Karyomeres, which are intermediate cleavage-stage structures with individual or groups of chromosomes enclosed by the nuclear envelope, failed to

**Fig. 6 | Functional analysis of novel genes. a** Two novel genes are identified from two WGCNA co-expression modules. *zeat1* has four TE-gene transcripts with insertion by a DNA transposon, Tc1N1_DR. *zeat2* has only one transcript without any TE interruption. **b** Expression patterns of two modules and two novel genes. Line plots showing the overall expression profiles (represented by the mean value of total genes in a certain module). The expression patterns of *zeat1* and *zeat2* are highlighted with red lines. The gray shades indicate the standard deviation. WISH experiment is used for validating the expression of *zeat1* and *zeat2*. **c** Morphological defects in M*zeat1* and MZ*zeat1* mutants at the shield stage (6 hpf). Both the number of embryos having the same phenotype as shown here and the total number of assayed embryos are listed at the bottom-right for each condition. Scale bar = 100 μm. **d** Assay of the *zeat1*-associated multinucleation. Interphase nuclei and nuclear membranes were labeled with H1m-GFP protein (green) and WGA-Alexa 647 (red) in WT and M*zeat1* mutant at the 64-cell stage (*n* = 3 independent experiments; 15 embryos were used in total). **e** Comparison of the cell cycle between the WT and M*zeat1* mutant embryos. Cell cycle time course from the 64-cell to 128-cell transition in WT and M*zeat1* mutants. Images were the projections of multiple confocal Z-slices. Four embryos were examined for each time point. **f** Morphology of WT and *zeat2*<sup>−/−</sup> mutant larvae (*n* = 20) at 3, 6, and 8 dpf (days post-fertilization). Scale bar = 100 μm. **g** Growth recording of *zeat2*<sup>−/−</sup> mutants. The numbers of larvae with different genotypes were recorded for a continuous 20-day inspection. The black arrow indicates the 18 dpf, when no *zeat2*<sup>−/−</sup> homozygous mutants can be detected. **h** The results of qRT-PCR. 18S rRNA was used as the internal control. Three biological replicates were used for each assay. Data are presented as mean values ± SEM. An unpaired *T*-test, two-sided was applied to determine statistical significance (insignificant results were not shown).

fuse properly, as seen in similar *bmb* maternal mutants[53]. The localization of *zeat1* to the nuclear envelope further supports its role in facilitating nuclear envelope fusion (Supplementary Fig. 19e). These results show that *zeat1* is a critical nuclear envelope-associated protein and is essential for karyomere fusion during early zebrafish embryogenesis.

The other novel gene *zeat2*, which was unannotated by Ensembl (version v103) at the time of data analysis and manuscript preparation but is currently reported as "*fastkd5*" in RefSeq release 227, was selected from the M2 co-expression module, where the genes have high abundance specifically at the 64-cell and 1k-cell stages and are functionally enriched in regulation of cell cycle, protein deubiquitination, and translation (Fig. 6a, b and Supplementary Fig. 19a). *zeat2* is a homolog of the *FASTKD5/Fastkd5* genes in humans and mice (Supplementary Fig. 19b), yet its function in zebrafish has not been reported. Zygotic CRISPR-Cas9 knockout mutants of *zeat2*<sup>−/−</sup> (Supplementary Fig. 19d) develop normally until 6 days post fertilization (dpf), whereafter showing a body curvature phenotype and apparent phenomenon of cell death (Fig. 6f). Finally, all the mutants cannot survive beyond 18 dpf, and therefore fail to develop into fertile adults (Fig. 6g and Supplementary Note 10).

To determine whether *zeat2* regulates non-canonical mitochondrial RNA processing like FASTKD5 in humans[54], we performed qRT-PCR to quantify the abundance of mitochondrial transcript precursors between WT and *zeat2*<sup>−/−</sup> mutant embryos. The zebrafish mitochondrial genome encodes 13 protein-coding genes, some of which have non-canonical junctions, including the ones in the 5′UTR of *co1*, the 3′UTR of *nd6*, and regions between *atp6* and *co3*, and *nd5* and *cyb*[54,55] (Supplementary Fig. 19f). We observed a significant accumulation of non-canonical precursor mRNAs in the absence of *zeat2*, with a particularly 27-fold enrichment of the 5′ UTR of *co1* (Fig. 6h). In contrast, canonical cleavage sites involving tRNA flanking, such as *nd1-nd2* and *nd3-nd4l*, showed no significant precursor accumulation in *zeat2*<sup>−/−</sup> mutants compared to WT (Fig. 6h). These results indicate that *zeat2* is an ortholog of *FASTKD5* and plays a similar role in processing mitochondrial mRNAs at non-canonical junctions. The loss of *zeat2* disrupts the processing of *co1*, *co3*, and other mitochondrial genes, which in turn probably affects the oxidative phosphorylation system and leads to developmental arrest in zebrafish embryos.

## Discussion

In contrast to the large size of genomic coverage by transposable elements, limited details of ZTA have been revealed in depth although TEs have been known to be activated[7,10,11]. Here we demonstrate that only a small proportion of TEs could be activated among the tremendous number of copies. It is worth noting that 550 TE-alone loci is a conservative estimate and several factors may influence the identification of active TE-alone loci, including stage-specific TE expression, low sequencing coverage, TE insertion polymorphisms among zebrafish strains and the TE annotation methods. Additionally, TE heterogeneity among zebrafish strains may contribute to a biased estimate of

TE-alone loci. The TE polymorphism between the experimental samples (AB and India strains) and the TU reference genome may lead to the misidentification of highly divergent TE loci and strain-specific copies.

By precisely identifying and quantifying the TE-alone loci/transcripts, we disclose the hierarchical heterogeneity of activation patterns among families, subfamilies, loci, and transcripts, and between TE-alone and TE-gene loci, which could be a result from their diverse sequence contexts and local epigenetics. These subtle but unique transcriptomic patterns of ZTA were largely mixed or masked in the previous studies until long-read sequencing is applied, and thus the development and application of the long read-based epigenetic assays will further benefit the understanding of ZTA[56–60]. While various types of TEs undergo polyadenylation[61,62], our analysis might overlook the detection of certain TE transcripts lacking poly(A) tails. Cytoplasmic polyadenylation prevalently existing in zebrafish early development may also lead to a bias in quantifying maternal TE transcripts[63,64] albeit significant underestimation is not observed for TE-alone transcripts between poly(A) enrichment-based and ribosomal RNA depletion-based RNA-seq (Supplementary Figs. 20 and 21 and Supplementary Note 11). This necessitates further investigation using a poly(A)-independent sequencing approach. To further reduce the impact on characterization of regular genes, we suggested to only consider the zygotically activated transcripts.

Despite that ZTA and ZGA are similar in several aspects, their difference is more remarkable, such as the highly heterogeneous trajectories and late activation time of ZTA. Indeed, limited resources in very early embryogenesis must be allocated to essential genes and proteins for cellular maintenance and zygotic genome activation. In parallel, previous studies showed that aberrant transposon accumulation at earlier stages can arrest ZGA and cause severe defects in embryogenesis[39,65]. A few TFs were previously reported to be involved in the TE activation (e.g., Dux, Klf5, Gata2)[32,66] and the TE silencing (e.g., KRAB-ZFPs)[67]. A recent study showed that a large ZNF gene family represses TE expression in zebrafish, underscoring a potential conserved approach in TE silencing as mammals[68]. In addition to *mxtx1/2*, we also analyzed 2,546 annotated zebrafish TFs[69] and found that 469 TFs were highly expressed with motifs identified at the activated TE-alone loci during ZTA (Supplementary Fig. 22 and Supplementary Note 12). This presumable machinery may be not fully supplied by maternal cells but would be only available until early ZGA is completed, because blocking the translation of earlier activated genes, such as *mxtx1/2* and other potential TFs can dramatically repress ZTA (Fig. 2f). In addition to the silence stage from zygote to cleavage and the activation stage from blastula to gastrula, we find an overall decreasing trend of TE-alone transcripts starting at gastrula and extending to pharyngula in zebrafish, with partial support at epigenetics (Fig. 5), which is also distinct from ZGA. The bulk-level data in the previous study also indicated the same decrease in several mammals[7] and prolonged TE transcription (i.e., LINE-1) across the normal repression process in mice can lead to developmental arrest[65]. Therefore, ZTA is

likely under a conserved control by multi-layer epigenetic regulation (Fig. 5).

TE-alone and TE-gene transcripts are more likely localized inside the nuclei based on both RNA-seq analysis and FISH experiments (Fig. 4). Although those TE-associated transcripts are significantly longer than regular gene transcripts (Supplementary Fig. 15e), which may potentially cause a longer time of transcription, the influences of the nascent TE-associated transcripts on the detection of subcellular localization remains unclear and requires further study. Our catalog of the full-length sequences of TE transcripts delivers informative clues to the persistent paradox concerning the prevalence of TE activation and the rarity of TE transposition. Most active TE-alone transcripts, such as ERVs that are the most abundant in zebrafish early embryogenesis, lack conserved protein domains (Fig. 1g), suggesting impaired transposition ability. In contrast, although most LINE transcripts possess intact coding regions (Fig. 1g), they are among the most nuclear-localized TE-alone transcripts (Fig. 4f and Supplementary Fig. 15c), indicating tight host genome control over their translation in the cytoplasm and thus autonomous transposition. Moreover, the PIWI-piRNA pathway likely adds another layer of post-transcriptional regulation upon the TE transcripts already transported into the cytoplasm before entering the translation machinery. However, transposition frequency is not zero although low, as most active TE-alone loci, especially LINEs and LTRs, are more likely originated by active transposition instead of segmental duplication (Supplementary Fig. 23a–d and Supplementary Note 13). This result suggests that some TEs can still be transposed without intact structure of domains. For example, an endogenous retrovirus in zebrafish, BHIKHARI, is likely actively transposed despite that it only encodes a Gag protein[14]. Additionally, the active TE-alone loci are predominantly harbored at genomic regions with relatively low recombination rates, such as sex chromosome[70] (Supplementary Fig. 23e), which may enhance their survival from genomic recombination elimination.

The full-length sequences are also informative for distinguishing TE-alone and TE-gene transcripts and enable reliable discovery of many unannotated loci/transcripts. While this study focuses on TE-alone transcripts, TE-gene chimeras represent a larger reservoir of TE-derived transcripts (total 25,082 versus 706; novel 20,330 versus 661) (Fig. 1b) activated during zebrafish early embryogenesis. The chimeric products, such as zeat1, may deliver the previously unknown functions and regulatory roles in zebrafish early embryogenesis (Fig. 6). The integration of different types of TEs with genes, along with the position and proportions of TE insertions, results in a more complicated stratification. Consequently, conducting another comprehensive, yet challenging, investigation of TE-gene transcripts is warranted to study how TEs cooperate with the host genome in embryogenesis.

## Methods

### Zebrafish embryo collection
Two zebrafish strains, AB and India WT, were used in this study. They were reared in a pH (7.2–7.6) controlled recirculation system at around 28 °C, with a 14-h light and 10-h dark cycle. Breeding involved placing males and females in a tank with a separator, followed by water replacement and separator removal for egg laying. Collected embryos, cultured in Holtfreter's solution (pH 7.0–7.4), were used for further generations or experiments. Eleven developmental stages of embryos were defined at specific time points, and embryos were collected at the following stages: fertilized egg (immediately after fertilization), 1-cell at 0.5 h post fertilization (hpf), 64-cell at 2, 1k-cell at 3, high at 3.3, oblong at 3.7, sphere at 4, dome at 4.3, 30%-epiboly at 4.7, 50%-epiboly at 5.3 and shield at 6 hpf, respectively.

### PacBio Iso-Seq library preparation and sequencing
Total RNA was isolated from each developmental stage of zebrafish embryos using TRIzol™ Reagent (Invitrogen, USA). RNA purity and concentration were assessed with the NanoPhotometer® Spectrophotometer (IMPLEN, CA, USA) and the Qubit® RNA Assay Kit in the Qubit® 3.0 Fluorometer (Life Technologies, CA, USA). The RNA integrity number (RIN) was determined using the RNA Nano 6000 Assay Kit and Agilent Bioanalyzer 2100 system (Agilent Technologies, CA, USA). RNA samples with a RIN ≥ 8 were used to synthesize cDNA with SMARTer™ PCR cDNA Synthesis Kit (Takara Bio, CA, USA). PCR amplification was performed using a KAPA HiFi PCR Kit (Kapa Biosystems, MA, USA) with the optimized number of cycles. Size selection of PCR products (cDNA) for each sample was applied using the Blue-Pippin System (Sage Science, MA, USA): <3 kb and >3 kb. Subsequently, two cDNA libraries (<3 kb, >3 kb) were prepared using a SMRTbell™ Template Prep Kit 1.0 (Pacific Biosciences, CA, USA) and sequenced on the PacBio sequel and sequel II platforms.

### Short read library preparation and sequencing
mRNA from five stages (fertilized egg, 1-cell, 64-cell, 1k-cell, and shield) was extracted with Dynabeads™ mRNA Purification Kit (Invitrogen, USA) and subjected to TURBOTM DNase (Invitrogen, USA) treatment. RNA-Seq libraries were prepared using the NEBNext® Ultra™ RNA Library Prep Kit for Illumina® (NEB, USA) and then sequenced at $2 \times 75$ bp paired-end mode on an Illumina HiSeq X Ten system with 8 replicates for each stage. Sequencing data quality was checked with FastQC (v0.11.8)[71] and the low-quality reads and adapters were removed using Cutadapt (v1.8.1)[72].

### Identification of TE-alone, TE-gene and gene transcripts
A total of 1.84 million CCS (Circular Consensus Sequencing) reads were extracted from PacBio subreads BAM files using ccs function embedded within Iso-Seq3 pipeline with "–all" parameter (https://github.com/PacificBiosciences/IsoSeq). The CCS reads were then divided into full-length (FL) and Non-full length (Non-FL) groups using BamTools (v2.5.1) (https://github.com/pezmaster31/bamtools) with the parameter "-tag rq = −1". All FL CCS reads were processed through the standard Iso-Seq3 pipeline, including adapter removal with lima, polyA tail removal with refine, and then clean reads were mapped on the zebrafish reference genome (GRCz11) with minimap2 (v2.24)[73] with the parameter "--MD -ax splice:hq -uf --secondary=no". The output alignment files in SAM format from all 11 developmental stages were fed together to TALON (v5.0)[74] for a combined transcript annotation with "--allowGenomic" parameter. We set the following criteria to filter out different types of unreliable transcripts annotated by TALON: (1) If one transcript is recorded in the zebrafish Ensembl reference annotation (v103), it is required to have ≥1 supporting read in at least one stage; (2) If one transcript is defined as novel transcript derived from a known gene locus, it is required to have at least 3 reads (equal to 1 TPM calculated based on our dataset) to support in at least one stage; (3) If one transcript is a novel transcript from an unannotated locus, it is required to have at least 2 reads to support in at least one stage. We also exclude extremely lowly expressed transcripts by filtering transcripts with <1 TPM at all 11 stages, except for autonomously expressed TE transcripts. We applied the same analysis with Non-FL CCS reads by TALON for read annotation, but Non-FL CCS reads are only used in quantification for transcripts identified by FL CCS reads.

All transcripts were classified into Known-Known (annotated gene locus and annotated transcript), Known-Novel (annotated gene locus and unannotated transcript) and Novel-Novel (unannotated gene locus and unannotated transcript) transcripts according to zebrafish Ensembl annotation.

We implemented CD-HIT (v4.8.1)[75] to merge incomplete and highly identical transcripts generated by TALON. We established specific criteria for clustering different types of transcripts defined by TALON, including: (1) Ensembl annotated transcripts were treated as true transcripts and not allowed to cluster, (2) autonomously expressed TE transcripts were not merged at this step and retained for further

manual annotation, (3) truncated transcripts, such as those annotated as ISM (Incomplete Splice Match) in known genes by TALON, were clustered at 98% identity, and (4) other types of transcripts, such as NIC (Novel In Catalog) and NNC (Novel Not in Catalog) were clustered at 100% identity to minimize the loss of potential novel transcripts.

For the final full-length transcript dataset, we quantified the expression abundance by combining both the FL and nonFL CCS read count for each transcript at each development stage. Transcript per million reads (TPM) was then calculated to represent the normalized expression value for each transcript. Gene-level expression level is the sum of all transcripts derived from the same locus.

### Classification of TE-alone, TE-gene and gene transcripts

Zebrafish TE consensus sequences were obtained from Dfam (v3.2) (https://www.dfam.org/) and used to annotate the latest released zebrafish reference genome (GRCz11) with RepeatMasker (v4.1.0) (https://www.repeatmasker.org). The TE and transcript annotation were compared based on genome coordinates, and all transcripts were classified into three categories: autonomously expressed TE transcripts (TE-alone), if the transcripts were embedded within TE regions over 90%, and the first exon was also covered by TE (TE promoter-driven transcription); TE-gene chimeric transcripts (TE-gene) if the transcripts had overlapped with TE annotation (≥10 bp, which is the smallest TE fragment in zebrafish reference genome), and gene transcripts, which had no overlaps with any TE-related sequences. TE-alone transcripts were then classified into different TE subfamilies, families, and types (DNA transposon, LTR, LINE, and SINE) according to the zebrafish TE classification system. TE-alone transcripts can be further separated into two subgroups based on the structure: transcripts from single-TE loci (singular) and chimeric TE-alone transcripts that are derived from multiple adjacent TE loci that belong to different subfamilies. Chimeric TE-alone transcripts can then be defined with major TE components (defined with subfamily name) and minor TE components based on the proportion of each TE within the transcript. The major TE components were used to classify chimeric TE-alone transcripts into different TE subfamilies, families, and types.

### Manual annotation of TE transcripts and genomic loci

By using Integrated Genome Viewer (IGV) (v2.16.0)[76], we compared gene annotation, TE annotation, TALON TE transcript annotation and read alignment together to manually check if the transcripts are truly transcribed from corresponding TE loci. A curated GTF annotation file recording the genomic coordinate information about TE loci and TE transcripts was created for further use.

### Comparison between short read and long read-based RNA-sequencing

Both Iso-Seq data (CCS reads) and Illumina RNA-seq data from mouse embryonic development were retrieved from NCBI (SRP225196) according to this study[28]. We applied the same pipeline for analyzing mouse long-read data as previously described for zebrafish, except for manual annotation for this initial comparison.

Clean reads from Illumina RNA-seq data were aligned onto zebrafish (GRCz11) and mouse (GRCm39) reference genomes by HISAT2 (v2.1.0)[77] with "-k = 1", respectively. Alignment results from each sample were then merged into a combined, sorted BAM file with SAMtools (v1.9)[78]. A combined transcript annotation was generated by using StringTie (v1.3.5)[79] and based on this annotation, sample-specific quantification was conducted by StringTie with "-e" parameter.

Both the number and the length of expressed TE-alone transcripts were compared between two sequencing technologies. Two BED files with expressed TE-alone loci annotation by two approaches were prepared to detect the overlapped TE-alone loci (detected by both long-read and short-read data) by *intersect* function in BEDTools (v2.30.0)[80].

A conventional approach was applied to define ZGA and ZTA during zebrafish embryogenesis with short-read RNA-seq data spanning the following five developmental stages: fertilized egg, 1-cell, 64-cell, 1k-cell, and shield. We first aligned short reads onto zebrafish genome using HISAT2 with default parameters and then quantified the gene/TE expression using StringTie with a combined annotation (Ensembl gene annotation and RepeatMasker annotation for zebrafish reference genome). Finally, we calculated the total TPM for all zygotic genes and TEs, respectively at each stage.

### LTR structure analysis

Based on our annotation, we defined the LTR locus as intact if it contains 5'LTR, internal coding sequence and 3'LTR structure. Losing more than one portion will be defined as truncated.

### Conserved domain analysis on TE-alone transcripts

Conserved domains for DNA transposons, LTRs, LINEs and ERVs were defined according to the previous description[81]. For TE-alone transcripts, open reading frames (ORFs) were predicted with TransDecoder (https://github.com/TransDecoder/TransDecoder) and protein sequences were generated to annotate conserved domains by HMMER (v3.3.1) with Pfam database (version 35.0)[82]. TE-related domains were collected and compared within different types of TEs.

### Define maternally deposited transcripts and zygotically activated transcripts

For TE-alone, TE-gene and gene transcripts, we defined a transcript as a maternal-deposited transcript if one transcript has TPM > 0 at either fertilized egg or the 1-cell stage. Otherwise, a transcript can be defined as a zygotically activated transcript.

### Whole mount in situ hybridization

For WISH experiment, we developed a specific probe design pipeline for TE subfamily. Simply, we extracted sequences for all transcripts from the target TE subfamily and aligned these sequences against all other transcripts (both TE-derived and gene-derived transcripts) according to our transcriptome annotation. Based on the alignment results, any overlapped sequences were detected and removed, and the remaining sequences were further processed with multiple sequence alignment by MAFFT (v7.455)[83] with default parameters. The consensus TE subfamily-specific sequences were extracted for WISH probe design.

The length of the probe for in situ hybridization was generally designed to be between 500–800 bp according to the length of unique regions, with the T7 promoter sequence directly added to the reverse primer of the probe. cDNA synthesized from reverse transcription was used as a template for PCR amplification of the target fragment. The PCR products were then checked by gel electrophoresis, and the desired bands were recovered from gel extraction. The recovered product is then used as a template for in vitro transcription to synthesize the probe, typically at 37 °C for 2 h.

Collect embryos at desired stages, and fix overnight in 4% paraformaldehyde at 4 °C. Next day, replace paraformaldehyde with PBS, then dehydrate using 50%, 70%, and 100% methanol for 5 min each at room temperature. Store at −20 °C for 2 h. For in situ hybridization, rehydrate embryos. Shift to RNase-free tubes and add DEPC + PBST gradually. Set hybridization temperature (55–65 °C). Pre-hybridization: 300 μl HYB−, then 300 μl HYB+ for 4 h + at the hybridization temperature. Add 1% probe diluted in HYB+, heat to 70 °C for 10 min, and add to the tube. Hybridize for 16 h at the hybridization temperature. Store retrieved probe at −20 °C. Clean by washing 50% formamide/2×SSCT (30 min), 2×SSCT (15 min), 0.2×SSCT (30 min) at hybridization temperature. Wash thrice with MABT for 5 min. Transfer to a 48-well plate, add substrate with levamisole, and stain at 28 °C. Monitor under a microscope, and stop reaction with 4% paraformaldehyde when desired.

## RNA-seq data analysis with zebrafish mutants and treated samples

We downloaded RNA-seq data for zebrafish embryo samples at 4, 6, and 8 hpf, including WT, α-Amanitin injection, cycloheximide treatment, triple LOF and mRNA rescue samples (PRJNA206070). Total RNA-seq data were first aligned onto zebrafish rRNA references to filter rRNA reads. Then, both poly(A) capture RNA-seq data and filtered total RNA-seq data were quantified using RSEM (v1.3.3)[84], with our long read-based sample-specific transcriptome reference. For each developmental stage and condition, the expression between WT and treatment was compared following $log_2(TPM + 1)$ transformation. Genes with low expression levels (TPM < 1) in WT samples were excluded from this analysis.

## Binding sites prediction of *mxtx1/2* on zebrafish ERV promoters

Quantification of *mxtx1/2* expression was performed using both short-read data (obtained from Expression Atlas on the EBI website) and long-read RNA-seq data from this study. The prediction of binding motifs of *mxtx1/2* was conducted with CIS-BP[85] (http://cisbp.ccbr. utoronto.ca/). As direct ChIP-seq data for *mxtx1/2* were not available in the database, binding motifs were indirectly inferred by considering closely related TFs in human and mouse. The ortholog relationship between Mxtx1/2 and DUX4 was established by OrthoFinder and PANTHER as the reciprocal best alignment (see https://www. alliancegenome.org/gene/ZFIN:ZDB-GENE-000710-7#orthology). For each ERV subfamily, the 5′LTR sequence was extracted, and potential binding TFs on these LTR sequences (promoter regions) were predicted using FIMO within the MEME suite[86]. Predicted binding motifs of *mxtx1/2* were subsequently compared with predicted TF motifs on ERV promoters to evaluate *mxtx1/2*'s binding potential on ERV elements.

A previously published ChIP-seq data of zebrafish Nanog and Mxtx2[34] were analyzed to identify genome-wide binding sites. The clean data were aligned onto the reference genome with Bowtie2 (v2.5.1)[87] and peak calling was performed using MACS2 (v2.2.9.1)[88] with default parameters. This resulted in 3534 and 33,111 peaks for Mxtx2 and Nanog, respectively. We further intersected these peaks with our TE annotation and found 43 Mxtx2 and 164 Nanog peaks overlapped with TE-alone loci. Of 43 Mxtx2 TE-related binding sites, 40 loci are ERVs and the other 3 are DNA transposons. However, Nanog can bind more diverse types of TEs, including 126 LTRs, 24 DNA transposons and 14 LINEs.

## TE activation during early mouse embryonic development

We utilized public Iso-Seq data (see above) covering mouse oocytes and embryos at the following developmental stages (1-cell, 2-cell, 4-cell, 8-cell, and blastocyst) for TE identification, with the same standard applied in zebrafish. In total, 28,939 full-length transcripts were identified and further classified into 20,321 gene transcripts, 7170 TE-gene chimeras and 1448 TE-alone transcripts.

Transcript quantification was performed using short-read RNA-seq data from the aforementioned stages. RSEM was employed with the established long-read transcripts as a reference, and with Bowtie 2 as the aligner with default parameters. Additionally, mouse single-cell embryo RNA-seq data was downloaded, spanning time points of 0, 2, 4, 6, 8, 10, and 12 hpf, from NCBI (PRJNA662943). Adapter sequences, along with six bases from the 5′ end and five bases from the 3′ end, were removed using Cutadapt (v1.18) (-u 6 -u -5). The clean data were used for transcript quantification using the same procedure. Transcripts Per Million (TPMs) were collected at each stage to create an expression matrix, and TPM < 1 at all stages were filtered out as lowly expressed transcripts. To quantify the abundance of each TE subfamily, TPMs were summed across all TE transcripts within the same subfamily. Mean TPMs were calculated from replicates at each respective time point. Maternally inherited and zygotically activated TE subfamilies were differentiated by setting a cutoff of TPM > 5 at the 1-cell or 0 hpf stages for maternally inherited TEs. Otherwise, these were categorized as zygotic TEs. To determine the activation stages of TE subfamilies, we required the expression at the previous stage to be <5 TPM, while at the tested stage ≥5 TPM and a continuous increase in at least two following stages.

## Examine the regulatory sequences of TE-alone loci from BHI-KHARI subfamily

We extracted 5′LTR sequences from all loci within BHIKHARI-I subfamily, conducted multiple sequence alignments using MAFFT. The structure and conserved motifs on 5′LTR sequences were identified based on a previous publication[35]. We also performed alignments between 5′LTR from BHIKHARI-I_DR and 5′LTR from BHIKHARI-3-I_DR and BHIKHARI-5-I_DR, and very low sequence identity was detected.

## Expression analysis of TE-gene and gene transcripts

Maternal and newly activated transcripts were defined for TE-gene and gene transcripts with the same criteria as TE-alone. For each stage, the total TPM and total number of activated transcripts were calculated. For TE-gene transcripts, we further divided them into five subgroups (0–20%, 20–40%, 40–60%, 60–80%, and 80–100%) according to TE proportion in the sequences.

## Expression analysis on family/subfamily/locus-levels

Locus-level expression abundance was calculated by summing the expression values of all transcripts derived from the loci. For expression analysis on the family and subfamily level, we calculated the sum of abundance for all TE-alone transcripts belonging to a certain TE family or subfamily.

## Expression trajectory analysis on the locus/transcript level

The R package "pheatmap" was used to perform unsupervised clustering and plot heatmaps. We used "Ward.D" as the clustering method and the clusters were identified using "cutree" method. Then, the median expression value of transcripts from each cluster was used to represent the average expression level. To estimate the expression complexity of TE-alone transcripts within each subfamily, we calculated the entropy of active TE-alone subfamilies. Specifically, we first calculated the number of TE-alone transcript expression patterns for each TE-alone subfamily. Then, the function "entropy" from the R package "entropy" was used to calculate the entropy for each TE-alone subfamily.

For TE subfamilies with multiple transcripts, we summarized the number of activation stages and expression clusters covered. For each TE subfamily, we determined the activation stage for each transcript and recorded the number of different stages this subfamily can cover. The same strategy was applied for summarizing the number of expression clusters one subfamily can distribute.

## Subcellular localization analysis with nuclear and cytosolic RNA-seq data

While direct comparison of RNA abundance between the nucleus and cytoplasm is not typically feasible[89], we can calculate the relative nucleus/cytoplasm (N/C) proportion to indirectly compare the subcellular localization of gene transcript, TE-gene transcripts, and TE-alone transcripts. A publicly available nuclear and cytosolic RNA-seq dataset[90], covering the 1k-cell, dome, and shield stages was downloaded (PRJNA599208), processed, and mapped onto our combined transcriptome. Both gene and transcript expression are quantified with RSEM. To compare the subcellular localization among TE-alone, TE-gene, and gene transcripts, we calculated a relative N/C proportion for each group at each stage. We first calculated the total abundance of TE-alone, TE-gene, and gene transcripts in the nucleus and cytoplasm, respectively; then we calculated the abundance proportion for

TE-alone, TE-gene, and gene transcripts in the nucleus and cytoplasm, respectively. Finally, the relative N/C proportion was measured by dividing the proportion in the nucleus by the proportion in the cytoplasm for each group. The same approach was applied to calculate the relative N/C proportion for each TE subfamily. Boxplots were created, depicting Q1 (25th percentile), Q3 (75th percentile), the maximum (Q3 + 1.5IQR), IQR (Interquartile Range), and the minimum (Q1 − 1.5IQR), with medians represented by center lines unless otherwise specified.

To test the prediction performance on subcellular localization with relative N/C proportion, we calculate relative N/C proportion with long intergenic non-coding RNAs (lincRNAs) and protein-coding genes at each stage, based on the prior knowledge that lincRNAs tend to localize inside the nuclei and protein-coding genes tend to localize in the cytoplasm. We first extracted all annotated lincRNAs and protein-coding genes according to zebrafish Ensembl annotation. Then, we removed maternally deposited lincRNAs and protein-coding genes and calculated the relative N/C proportion following the approach previously described.

### Fluorescence in situ hybridization

Candidate TE subfamilies were selected for subcellular localization validation using FISH. Representative TE subfamilies with moderate to high expression levels and different relative N/C proportion from DNA transposons, LTR and LINEs were used for experimental validation. Specific probes for each candidate subfamily were designed with the same strategy as WISH probe design described above.

Embryos were prepared according to the WISH protocol. Sequentially, embryos underwent hybridization rounds: Anti-digoxigenin antibody (Digoxigenin-AP antibody, Catalog#: 11093274910, Roche) (POD-conjugated) was diluted 1:1000 in blocking solution and incubated overnight at 4 °C with rotation. Post-incubation, samples were switched to MABT solution and washed thrice for 30 min at room temperature. Subsequently, a single 5-min wash with PBST was performed. For signal enhancement, Cy3-labeled tyramide reagent (PerkinElmer, 1:50 in amplification dilution buffer) was applied to embryos, followed by a 20-min dark incubation and a 10-min PBST wash. Blocking involved 1% BSA and 10% inactivated goat serum in PBST at room temperature for 1 h. β-catenin (L54E2) mouse antibody (1:200) (Catalog#: 2677, Cell Signaling Technology) was applied and incubated overnight at 4 °C. Afterward, samples underwent three 15-min PBST washes. A secondary antibody (Alexa Fluor 488 AffiniPure Goat anti-Mouse IgG, Catalog#: 115-545-003, Jackson ImmunoResearch Labs) was applied and incubated at room temperature for 2 h. Post-secondary antibody incubation, DAPI (0.5 μg/ml) was used for a 10-min incubation at room temperature. The samples were then subjected to three 15-min PBST washes. Following PBST removal, samples were mounted with an anti-fluorescence quenching mounting agent suitable for confocal imaging.

### Short-read RNA-seq quantification of TE-alone loci beyond the shield stage

A benchmark RNA-seq dataset over the early development of zebrafish was downloaded from European Nucleotide Archive (PRJEB7244, PRJEB12296 and PRJEB12982)[19]. For each developmental stage, we quantified the expression of TE-alone transcripts using RSEM (v1.3.3) based on our long read-established transcriptome.

### Epigenetic regulation on activation and repression of TE-alone loci

ATAC-seq data were downloaded from NCBI (GSE130944)[42] and aligned onto the genome with Bowtie2 (v2.5.1)[87] with parameters "--very-sensitive –no-mixed –no-unal -X 2000". PCR duplicates were removed with "MarkDuplicates" function in GATK4. Uniquely mapped reads were extracted with SAMtools with "-q 30". BAM files were converted into "bw" files with "bamCoverage" function in deep-Tools (v2.0)[91].

H3k4me3 and H3K27ac data were downloaded from NCBI (PRJNA473799 and PRJNA434216) and analyzed with the same method used for ATAC-seq data.

H3K9me3 ChIP-seq data were downloaded from NCBI (PRJNA449956)[44] and mapped onto genome with Bowtie (v1.3.1)[92] with "-M 1 –best –strate". PCR duplicates were marked and removed with MarkDuplicates. ChIP and control BAM files were converted into "bw" file with "bamCompare" function in deepTools.

Small RNA data were downloaded from NCBI (PRJNA215266)[45] and were aligned using the same strategy as ChIP-seq data. BAM files were converted into a "bw" file with "bamCoverage" function in deepTools. Besides mapping onto the genome, we also aligned the data onto our TE-alone transcripts.

Heatmap visualization for the above datasets was generated by "computeMatrix" and "plotHeatmap" functions in deepTools with default parameters. BED files recording 5' TSS to 3' TTS used for deepTools were generated according to our transcriptome annotation.

We downloaded zebrafish piRNA collection from piRBase Release (v3.0)[93]. Small RNA reads mapped onto TE-alone regions were extracted and aligned against the zebrafish piRNA database using Bowtie (v1.3.1) with default parameters. The percentage of annotated and unannotated reads was summarized for each stage. We investigated several sequence features of piRNAs, including the total number of mapped reads, read length distribution, and the 5' uridine and adenine at the tenth base. Sequence logos were created using WebLogo 3 (ref. 94).

"piRNA" as a keyword was used to search for the relevant gene names on The Zebrafish Information Network (ZFIN) (https://zfin.org). Genes involved with piRNA-dependent silencing, including *piwil1* and *piwil2*, were recorded and quantified with short-read RNA-seq data previously described.

### Functional analysis of the unannotated genes

We combined the transcripts from novel gene and TE-gene loci and prepared sequences for 2895 novel transcripts for several downstream analyses. First, we predicted the coding potential for these transcripts using CPC[95] (https://cpc.gao-lab.org/), which resulted in 1622 protein-coding and 1273 non-coding RNAs. Second, we conducted functional analysis on the 1622 protein-coding mRNAs. We extracted the protein sequences of the longest mRNA to represent each gene and ran InterProScan (v5.61-93.0)[96] to search the Pfam domains. In addition, we aligned protein sequences of all novel genes against all protein sequences of human and zebrafish and identified 367 genes with homologs in both or either species.

### TE-related TF prediction

To computationally predict the potential TFs in regulating TE expression or repression, we first ran FIMO function within the MEME Suite to search the known motifs for 550 TE loci and identified 486 TFs with potential binding sites ($p \leq 0.00001$) on TE's promoter regions. We then investigated the expression pattern of 2546 annotated TFs (downloaded from AnimalTFDB 4.0)[69] during early embryonic development in zebrafish and found that 2120 TFs have expression ≥1 TPM at ≥1 stage. Of 486 potential TE-regulating TFs, 469 are highly expressed during early development in zebrafish. A heatmap was generated to illustrate the expression pattern of these TFs.

### The comparison of RNA-seq data generated by different library preparations

Two publicly available RNA-seq datasets (PRJNA624126 and PRJNA529241) covering the zebrafish early development generated by both poly(A) capture and ribosomal RNA depletion library preparation

methods were downloaded and quantified using Salmon (v1.10.0)[97]. The TPMs estimated by different library preparation methods were compared and displayed in scatter plots for each transcript. For the data with biological replicates (PRJNA529241), we further applied DESeq2 (Log$_2$FC > 2 & FDR < 0.01) to identify significantly differentially expressed genes between two methods.

## Evolutionary analysis on TE-alone loci

Kimura divergence (KD) values for each copy were retrieved from RepeatMaker results. Among DNA transposons, LTRs and LINEs, we compared the KD distribution between active and silent copies. For LTRs, we also compared the KD distribution between active intact and silent intact copies. Truncated and intact TE copies are determined by the sequence comparison with consensus sequences for the corresponding subfamily. We required the intact copy to account for at least 80% of consensus sequences and 5′end loss of no more than 30 bp and 3′ end loss no more than 100 bp. For each stage, Pearson's correlation coefficients were calculated between KD and TPM.

A de novo LTR annotation was conducted with LTR_retriever[98]. The insertion years of intact LTRs were calculated based on the sequence divergence of two LTRs with $T = K/2\mu$, where $K$ and $\mu$ are the divergence rate and neutral mutation rate. The $K$ is estimated by the Jukes-Cantor model (1969) with $K = -3/4*\ln(1 - d*4/3)$, where the proportion of sequence differences, $d = 100\%$-identity%, and a neutral rate of $1.46 \times 10^{-8}$ in fish genomic evolution was used[99]. We compared this de novo annotated LTR dataset with our expressed LTR annotation to find the common copies between the two datasets. The density profiles of evolutionary ages for both activated LTR and silent LTR groups were drawn and compared with the Kolmogorov–Smirnov test.

We downloaded genome assemblies for additional three zebrafish strains, including AB (PRJEB38589), Nadia (PRJEB38577) and CB (PRJEB38573) from NCBI for comparative genomic analysis. To identify the potential orthologous regions of expressed TE-alone loci on TU reference from these three strains, we extracted 25 kb upstream and downstream of TE-alone loci and aligned the sequences (50 kb surrounding sequences plus TE-alone sequences) against AB, Nadia and CB genomes with BLASTN (v2.13.0)[100]. The best-aligned regions were identified, extracted and then validated by "nucmer" function within MUMMER (v3.23)[101]. Dot plots were generated by "mummerplot" function within MUMMER. The presence and absence of TE-alone loci on each genome were assayed and assigned to corresponding evolutionary splits across the phylogeny of four zebrafish strains modified according to the previous study[102]. TE-alone loci were then classified into four major groups according to this comparative analysis and evolutionary relationship among strains: TU (TE-alone loci existing in only TU genome), TU-AB (presence in TU and AB genomes), TU-AB-Nadia (presence in TU, AB and Nadia genomes) and TU-AB-Nadia-CB (presence in all four genomes), from evolutionarily young to old. Further, we combined TU and TU-AB into the young group and TU-AB-Nadia and TU-AB-Nadia-CB into the old group for comparison on ZTA pattern.

For four subgroups formed based on the evolutionary relationship among TU, AB, Nadia and CB, we tested the relationship between TPM and evolutionary ages. The activation and expression patterns between evolutionarily old and young groups were also tested. We calculated the number of TE-loci activated in each of the eight developmental stages and distributed in each of the eight clusters we established based on expression.

To determine which active TE-alone loci were originated by TE transposition or segmental duplication, we assumed that segmental duplication would copy both the TE loci and flanking regions, while transposition may only duplicate TE loci and perhaps a small number of flanking bases. Therefore, by comparing both the sequences from TE-alone loci and flanking regions between the target TE-alone locus and corresponding homologous region, we can differentiate these two mechanisms: if both TE-alone loci and flanking regions show high similarity, this TE locus is more likely originated by a segmental duplication event; otherwise, it is more like a transposition.

To do so, we first conducted genome-wide searching for homologous regions of each TE-alone locus, by employing minimap2 ($N = 1000$) to output all aligned regions. From this alignment result, the best alignment was evaluated and selected as the potential donor for a given TE-alone locus, which will be used to further infer the origin mechanism. For a pair of TE-alone locus and an identified donor region, we extracted the TE sequences plus extension into the flanking regions from 1 bp to 10 kb and conducted pair-wise alignment using MUMMER. If both the TE locus and its two flanking regions show high identity, the origin mechanism would be classified as segmental duplication; otherwise, it was categorized as TE transposition. We used 500 bp flanking regions (both 5′ and 3′ ends of TE-alone locus) as a cutoff to calculate the proportion of these two mechanisms in contributing to active TE-alone loci evolution. For DNA transposon, LTR and LINE, we employed the Chi-squared test to investigate if there was a different preference in either segmental duplication or transposition as an amplification approach.

## Weighted correlation network analysis

We utilized the WGCNA (v1.72.1) package[103] in R (v4.2.3) to perform co-expression network analysis on 40 short-read RNA-seq samples from five stages (fertilized egg, 1-cell, 64-cell, 1k-cell and shield). The short-read RNA-seq data were quantified with kallisto (v0.46.0)[104], using our long read-based transcriptome annotation established above. Differential expression analysis between two neighboring stages was conducted with DESeq2 (v1.38.3)[105] using the filtering criteria of log$_2$(Fold Change) ≥ 1 and FDR ≤ 0.01. The resulting differentially expressed genes were used to establish a co-expression network with the "blockwiseModules" function in the WGCNA package. A pair-wise correlation matrix was computed for each set of genes, and an adjacency matrix was calculated by raising the correlation matrix to a power of 12, which was selected based on the scale-free topology criterion for network construction. The co-expression gene networks and modules were visualized and established with Cytoscape (v3.5.1)[106].

## Gene ontology enrichment analysis

Gene sets were assayed with DAVID[107] (https://david.ncifcrf.gov/tools.jsp) for functional enrichment analysis. The DAVID results were downloaded and processed for dot-plot display with R (v4.2.3).

## Homologous gene analysis

Gene domains were predicted by NCBI Conserved Domain Search Tools (https://www.ncbi.nlm.nih.gov/Structure/cdd/wrpsb.cgi)[108]. Homologous genes were retrieved from NCBI by BLASTP with protein sequences of query genes[100].

## Constructing phylogenetic trees

Multiple sequence alignments were conducted with protein sequences from homologous genes using MAFFT. Phylogenetic trees were established using FastTree (v2.1.11)[109] and drawn using iTOL (https://itol.embl.de/upload.cgi)[110].

## Design gRNA for CRISPR-Cas9 KO

CRISPR-Cas9-mediated mutagenesis was performed as previously reported[111]. The guide RNA (gRNA) target site was selected with CHOPCHOP (v3)[112]. The gRNA sequences used in this study were listed as follow:

*zeat1*: 5′-TAATACGACTCACTATAGggacccgctgcagctccaggGTTTTAGAGCTAGAAATAGC-3′.

*zeat2*: 5′-TAATACGACTCACTATAGggaccatctcccatggttggGTTTTAGAGCTAGAAATAGC-3′.

Each gRNA sequence contains the T7 promoter (upper case), a specific DNA-binding sequence (lower case), and a constant 19-nt tail

(upper case). The gene-specific sequence was used in combination with the reverse universal sequence (5′-AAAAAAAGCACC-GACTCGGTGCCAC-3′) to amplify the target sequence from the corresponding DNA.

## Target validity detection and $F_0$ preparation

gRNA and Cas9 protein were diluted in nuclear-free water, gRNA at 50–100 ng/µl and Cas9 at 50 ng/µl. The mixture was microinjected into single-cell stage zebrafish embryos (50 injected, 10 as non-injected controls). After 24 h, 5 developing embryos were lysed in 50 µl 50 mM NaOH at 95 °C for 30 min, neutralized with 5 µl 1 M Tris-HCl (pH 8.0), and used for PCR assay. Gel electrophoresis was used to confirm PCR product size. T7E1 enzyme cleavage was applied to assess target efficiency. Positive PCR products were then sequenced. Upon confirmation, large-scale injections (around 200 embryos) yielded $F_0$ fish.

## Mutant screening and propagation

$F_0$ zebrafish were bred with WT adults. $F_1$ embryos were collected at 24 h post-fertilization, with eight embryos per pair tested for mutations. Detected mutations led to the rearing of the remaining $F_1$ embryos. Upon maturity, $F_1$ fish underwent caudal fin clipping for mutation genotyping. Selected heterozygous $F_1$ adults were crossed to produce $F_2$ embryos, nurtured for a month, and genotyped. $F_2$ homozygous mutants produced $F_3$ embryos. Maternal mutants arose from $F_2$ homozygous females crossed with WT males, while maternal-zygotic mutants arose from the self-crossing of $F_2$ homozygous mutants. This method generated the heterozygous or maternal mutants discussed in this study.

## Live imaging

One-cell embryos were injected with h1m-GFP protein (green) and WGA-Alexa 647 (red) and manually dechorionated. Embryos at the 64- to 128-cell stages were mounted in 5% methylcellulose, and images were taken on Leica TCS SP8 microscope in an environmentally controlled chamber (26 °C).

## qRT-PCR

The precursor transcript abundance of mitochondrial genes and pre-processed junctions were measured on RNA isolated from 10-dpf zebrafish using the miRNeasy RNA extraction kit (Qiagen). cDNA was prepared using reverse transcriptase (Promega) with random hexamers and used as a template in the subsequent PCR that was performed using a Roche Light Cycler 480 machine and analyzed following the $2^{-\Delta\Delta\text{che}}$ method. 18S rRNA was used as the reference gene for qRT-PCR. An unpaired $T$-test was performed with Prism 10.

## Statistics and reproducibility

No statistical method was used to predetermine sample size. No data were excluded from the analyses. For long read RNA-seq data, no biological replicates were used, but for short-read RNA-seq data, eight biological replicates were used to generate the data. Unless specifically stated, a boxplot shows the center as median, lower bound of the box as the first quartile (Q1), upper bound of the box as the third quartile (Q3) and lower whisker (Q1 − 1.5 × IQR) and upper whisker (Q3 + 1.5 IQR) as minima and maxima.

## Reporting summary

Further information on research design is available in the Nature Portfolio Reporting Summary linked to this article.

## Data availability

All original sequencing data generated in this study have been submitted to the NCBI Sequence Read Archive (SRA) and are accessible under the BioProject accession number PRJNA1028258. The previously published data with the detailed description used in this study is listed in Supplementary Data. A video presentation introducing this project is available on the website (https://youtu.be/Bb-qBZZNQo4). Source data are provided with this paper.

## Code availability

The code of the major analysis modules in this study (including data quality control, alignment, transcript identification and quantification, TE-derived transcript identification and classification) are packaged into a bioinformatics pipeline called aTEA. aTEA is publicly available at https://github.com/Augroup/aTEA.

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

## Acknowledgements

This work is supported by an institutional fund from the Gilbert S. Omenn Department of Computational Medicine and Bioinformatics, University

of Michigan (K.F.A., B.L., D.W., Y.W., and P.T.) and an institutional fund from the Department of Biomedical Informatics, The Ohio State University (K.F.A., B.L., D.W., and Y.W.). We thank the computational service provided by the Ohio Supercomputer Center and Advanced Research Computing, University of Michigan. S.J. acknowledges the fund from the National Key Research and Development Program (2024YFA1803000 and 2024YFA1802200) and the National Natural Science Foundation of China (92254302 and 32293202), and Y.G.Y. acknowledges the fund from the National Natural Science Foundation of China (32121001). We thank Dr. Aifu Li for the helpful discussion and proofreading.

## Author contributions

K.F.A. and S.J. designed and supervised the study. B.L. and D.W. analyzed the data. T.L. and Y.Y. performed experiments. B.L., T.L., P.T., and Y.W. visualized the results. B.L., K.F.A., and S.J. wrote the manuscript with assistance from T.L., D.W., Y.Y., P.T., Y.W., and Y.G.Y.

## Competing interests

The authors declare no competing interests.

## Ethical approval

All fish experiments in this study were approved by the animal care and use committee of the Institute of Genetics and Developmental Biology, Chinese Academy of Sciences.
