## [Transparent Peer Review file · Nature Communications]

Zygotic activation of transposable elements during zebrafish early embryogenesis

Corresponding Author: Professor Kin Fai Au

Version 0:

Reviewer comments:

Reviewer #1

(Remarks to the Author)

In this manuscript Li and colleagues discover the TEs that are expressed during early zebrafish development. They perform long-read sequencing at 11 stages of early zebrafish development and analyze the specific TE copies that are expressed either as TE-alone or as part of the gene TE-gene. They find two waves of TE expression after ZGA and this was not corroborated by expression analysis using short read sequencing. They also break down the analysis to TE families and subfamilies. They touch upon TE regulation by analyzing some histone modifications and they also analyze the evolutionary aged of TEs and find that younger TEs are the ones that are mostly expressed. Most of the TE transcripts are nuclear. Finally, they do some experiments with two new loci or two new TE-genes that they identify and show that they are important for embryonic development.

In my opinion the analysis of TEs in this manuscript is very thorough and the paper has merit as a resource for the community even though its biological aspects may be more limited.

1. The authors should deepen their investigation on the regulations of TE expression. They mostly work on their repression but what happens to their activation. What is the correlation with active histone marks?
2. Are there any specific transcription factors that regulate TE expression or repression? Can the author perform motif analysis?
3. Can the authors discuss what is the potential functions of the TEs? What is known about TEs in early zebrafish development but also in the development of different organs?
4. The authors could describe better the newly found genes
5. It is not obvious from the text what is new data and what is reanalyzed data
6. What is the function of *zeat1* and *2*. A brief description is provided but the authors don't dig deeper to the function of these chimeric genes

(Remarks on code availability)

Reviewer #2

(Remarks to the Author)

This manuscript reports findings on the activation of transposable element (TE) expression during the early stages of zebrafish embryogenesis. Using long-read sequencing, the authors are able to resolve TE expression at the locus level, and find that only a small subset of TE loci produce transcripts (as opposed to chimeric gene-TE transcripts). They demonstrate that zygotic TE activation (ZTA) follows zygotic gene activation (ZGA), but apparently also proceeds in two waves. I enjoyed reading this manuscript and thought there were several interesting analyses. The discovery of ZTA, temporally distinct from ZGA, is novel and has (to my knowledge) not been reported before, although as the authors note, future studies with long-read sequencing will be needed to determine whether this is specific to zebrafish or a general feature of other animals. This paper will be an important reference for people working in zebrafish embryology, and further advances the use of this organism as a model for studying the activity of TEs in early development. Overall, it is clear that the authors have put a lot of effort into this project, and the experiments are clearly described and appear to have been rigorously executed. I have

only one major concern, and am certain that this can be addressed without significantly changing the manuscript, but will possibly some re-writing and cutting of small sections. Other than that, I have several other suggestions to improve the manuscript, most of which can hopefully be addressed without the need for further analyses. Feel free to concentrate on the ones you feel are most important and ignore some others.

Major concern:

1. An important note about poly-A enrichment methods and maternally deposited transcripts: This is one aspect of the paper that does require significant revision, or at least a more thorough disclaimer for interpreting results. In animals (and certainly zebrafish: <https://doi.org/10.1242/dev.159566>), maternally deposited transcripts undergo cytoplasmic polyadenylation, and thus are initially much harder to detect via poly-A enrichment-based sequencing platforms. The authors do briefly mention polyadenylation in the first paragraph of the discussion, but I think this is a bigger issue than that, as it could lead to dramatic underestimation in the number of maternally deposited transcripts, and skewing of the trajectories described in fig 3d.

Minor concerns/suggestions:

2. The authors identify 706 TE-alone transcripts deriving from 550 distinct loci. Given the abundance and diversity of young TE families in the zebrafish genome, this is a remarkably small number. It would be useful to get an idea of how conservative this estimate is, and the degree to which it is influenced by the thresholds used for defining “TE-alone”, or methodological aspects such as sequencing depth. In particular, I think it is important to briefly point out that the approach does not take into account TE insertion polymorphisms. Zebrafish are known to be highly heterozygous, and there are certainly many polymorphic TE loci between individuals. Thus, it is possible (even likely) that some TE-alone transcripts are discarded because they arise from loci that exist in the sequenced individuals, but not the reference genome.

3. The authors use the term “active” to describe TEs throughout the manuscript. It would be good to briefly clarify somewhere in the introduction or first results paragraph that you are referring to transcriptional activity, not necessarily transpositional activity.

4. I thought the analysis of core domain losses in TE-alone transcripts was interesting. While this does suggest that many families are incapable of autonomous transposition, I think it would be good to emphasize that they are transposing nonetheless! This is clear from the fact that several TE families have many full-length, nearly identical copies in the genome, despite lacking many of the core domains required for fully autonomous transposition. For example, BHIKHARI encodes only a single Gag protein, and yet is very likely still actively transposing in zebrafish populations ([doi:10.1101/2024.03.25.586437v1](https://doi.org/10.1101/2024.03.25.586437v1)).

5. With respect to the timing of ZTA, I agree with the authors’ suggestion that it likely relies on host factors provided by ZGA, thus explaining the relative delay of ZTA. However, I am more skeptical about the existence of two distinct ZTA waves. Do the authors reach this conclusion solely based on the correlation analysis shown in Fig 2e? Are there consistent differences between the TEs activated at 1k versus oblong stage? E.g. retroelements in the former, DNA transposons in the latter? Is there any evidence to suggest that major and minor waves of TE activation would be conserved in other species or is this just a quirk of the TE composition in zebrafish.

6. The authors perform a detailed analysis of TE-alone transcript trajectories in Fig. 3. I liked this section but would be cautious about over-interpreting the differences between the categories described in Fig. 3D. Many of these patterns look qualitatively similar to one another, if allowing for some noise and variation. Focusing on BHIKHARI again, the authors report that the transcripts can be separated into six different trajectory types. However, as the authors note, the regulatory regions of BHIKHARI are highly similar to one another, and have clearly conserved regulatory elements. It would therefore be surprising if the expression patterns really did fall into distinct groups, as opposed to being an artifact of the threshold chosen for the hierarchical clustering step. From looking at the dendrogram in Fig 3D, it looks like 3 or 4 clusters might be a more natural cut point, and it would be interesting to see if this modification would lead to more BHIKHARI transcripts falling within the same cluster.

The authors attribute the intra-family heterogeneity in trajectory types to the “possible existence of unknown regulatory sequence variations or epigenetics layers”, and I agree with this, but think it might be overstating the case – we already know that different TEs (particularly ERVs/LTRs) recruit distinct sets of transcription factors, and as for epigenetic layers, since TEs insert (mostly) randomly into the genome, it is natural that there would be variation in transcriptional strength and pattern due to the varying genomic contexts that they find themselves in.

7. I liked the analysis of subcellular localization differences, but think that the interpretation could be improved. The finding that DNA transposons tend to have lower N/C ratios than LTRs/LINEs is consistent with known mechanisms of replication – since DNA transposons do not use an mRNA template to replicate, they have no need to transport transcripts to the nucleus. LINEs replicate through target-primed reverse transcription, and thus explicitly need to transport their mRNA template to the nucleus, and should thus have high N/C ratios. However, it is somewhat surprising that LTRs have high N/C ratios, since reverse transcription takes place in the Gag capsid in the cytoplasm (as far as we currently understand), and the mRNA template is degraded in the process. Thus, finding high levels of LTR mRNA in the nucleus might suggest incomplete reverse transcription and unintended import of mRNA rather than cDNA. Incidentally, the fact that LTRs and LINEs are enriched in the nucleus at all is strong circumstantial evidence for active transposition in zebrafish!

8. Since the authors discuss H3K9me3 and piRNAs, they might be interested to know that a large family of zinc fingers proteins (FINZ-zinc fingers) have recently been shown to be repressing TE activity in zebrafish, likely through recruiting H3K9me3 to TE loci, as is the case for KRAB-zinc fingers in mammals ([doi: 10.1101/gr.277966.123](https://doi.org/10.1101/gr.277966.123)).

I hope these comments are of use to you!

Kind regards,
Jon Wells

(Remarks on code availability)

I did not run the code due to the need to the significant number of software and data dependencies. However, the codebase is logically organized, and the code is readable and well-commented. I am confident that it is high-quality and could be re-used if so desired.

That being said, I couldn't find scripts or notebooks used for generating figures, which is a shame since they are very attractive! It would be helpful to make these available in the github repo too.

Reviewer #3

(Remarks to the Author)

Li, Li, Wang et al present a detailed analysis of transposable element (TE) expression during the zebrafish maternal-to-zygotic transition. Their use of long-read RNA-sequencing along with conventional short-read RNA-seq expands in a tight timecourse reveals insights into the temporal dynamics of TE activation, thus expanding on the previous work of Chang et al 2022 that originally mapped TE expression across zebrafish development. For this reason, this paper is a solid contribution to the field. However, the manuscript as written overreaches somewhat in the novelty and significance of their findings, especially regarding the functional consequences of TE expression in the early embryo, and in revision, care should be taken to frame these results appropriately.

Major points:

- The expression patterns as exemplified in Fig 2a lead the authors to define and coin the term "Zygotic TE activation" (ZTA) along with distinguishing a minor and major wave of ZTA. It is difficult to see how TE activation patterns differ sufficiently from activation of other lowly expressed genes to warrant defining a "ZTA." Similarly, the inflection point in expression increase that the authors use to distinguish the minor from major ZTA does not convincingly implicate two waves. Different genes have different activation kinetics, even within a wave, owing to their unique regulatory contexts. There is just not enough evidence that TEs as a group have a unique regulatory profile. The authors use of a-Amanitin, pluripotency factor LOF, and cycloheximide RNA-seq data to justify the existence of ZTA all just seem to show that expressed TEs behave like other genes during ZGA. The mxtx1/2 analysis in Fig 2g / Supp Fig 9 in particular seems to suggest that a lot of ERV expression is *not* downstream mxtx1/2. By contrast, the minor and major waves of zebrafish zygotic genome activation have strong mechanistic foundations (see, e.g., Hadzhiev et al, Dev Cell 2023). The concept of a distinct ZTA is as yet speculation, and thus should be limited to the discussion.

- It is unclear how the CRISPR LOF analyses demonstrate the functional significance of TEs per se. The TC1 element in "zeat1" seems to be occurring in an extended 3'UTR according to the RefSeq track on the UCSC genome browser (so, not really a chimera, and misleading figure Fig 6a). Given that a lesion in the CDS was generated, the phenotype is almost certainly not due to loss of the TE but rather loss of the encoded protein. Regardless, demonstrating an injected mRNA rescue would strengthen the phenotypic claim. For "zeat2", Fig 6b shows a 932bp region, which is at odds with the protein size suggested in Supp Fig 19b; it has no annotated refseq or ensembl gene model, nor does there appear to be any obvious RNA-seq expression evidence. So, further clarification is needed for what the zeat2 gene is.

- The similarities in ERV activation between zebrafish and mammals is absolutely notable; however, the direct analogy to DUX4/Dux, which have roles in ZGA and downstream TE activation, is a big stretch. How the authors concluded that the zebrafish ERVs are orthologs to DUX4/Dux needs to be clarified, since DUX4 and Dux are not even unambiguously orthologs of each other (they have been called "functional orthologs" despite almost completely dissimilar sequence, suggesting that they actually have convergent rather than conserved functions). The authors should revisit how they discuss these observations.

- It's unclear to what extent the nuclear localization conclusions may be influenced by signal from nascent transcripts. Are TE-associated transcripts longer on average (and thus, slower transcribed)? Do the FISH images look similarly nuclear at later stages? Why is a bulk all-gene "relative N/C ratio" being calculated as opposed to a nuclear/total value for each gene -- could it be that with so many non TE-genes, any compartment bias is just averaged away?

- Concordance (or lack thereof) with the Chang et al Genome Res 2022 study should be made more explicit. E.g., the Chang study finds many more maternally provided TE associated transcripts -- is the discrepancy due to how TEs are defined?

- The methods suggest that both AB and a wild strain (is this WIK?) were used to generate the transcriptomics, which we might expect could have divergent TE expression patterns. However, it does not seem that any of the paper's analyses take this into account or acknowledge the possible TE heterogeneity in the collected data (this would seem to complicate the analysis in Supp Fig 14, e.g.). The authors should clarify

Additional points:

- "NC ratio" has a very specific usage to refer to volumes, not RNA quantities, especially in the context of genome activation.

The authors should find an alternate way to name this value (or, simply use nuclear proportion instead)

- Fig 4a, the TN and TC variables seem to be used both for TEs and for genes

- Xu et al, Dev Cell 2012 indeed has mxtx2 ChIP-seq data, which could be useful for the authors (cf Methods section 13)

- Probe sequences for the in situ experiments should be provided

(Remarks on code availability)

The github repository would've required me to download and expand a tar-ball onto my computer, which I was unwilling to do. As far as I can tell, most of the functionality is likely running pre-existing bioinformatics software, with the exception of how the transposable elements were sorted into different groups. The README seems appropriately detailed

Version 1:

Reviewer comments:

Reviewer #1

(Remarks to the Author)

The authors have adequately answered all my points.

(Remarks on code availability)

Reviewer #2

(Remarks to the Author)

The authors went above and beyond in their revisions to this paper. All my concerns were satisfactorily addressed and I have no further comments.

(Remarks on code availability)

The code is complete and easy to read. I have no further issues with it.

Reviewer #3

(Remarks to the Author)

In revision, the authors put a lot of effort into adding additional experiments and analyses to support their claims. Unfortunately, I do not think these efforts have been adequate relative to how the paper is presented. Although this study still provides the field with potentially useful raw data in the long-read sequencing and characterization of TE expression dynamics, their major claims of biological significance are unwarranted at this time: namely, (1) the data are still insufficient to support the existence of a distinct event of TE activation separate from ZGA, and (2) the phenotypes associated with the mutants cannot as yet be attributed to TEs. Especially regarding the second point, the manuscript seems to be written in a way that downplays and/or neglects important details that are necessary for a full, objective interpretation of the results, which is certainly not ideal for the reader.

Main issues:

- The interpretation of the mutants and their phenotypes remains problematic relative to the potential roles of TE-gene fusions. The TE in "zeat1" is indeed in the UTR (which the authors still do not make clear in either the main text or Fig 6a; they bury this fact in the supplement), so the phenotype of the LOF mutation that disrupted the ORF would by default be attributed to loss of the protein unless demonstrated otherwise. "zeat2" is indeed annotated by RefSeq as fastkd5, and the ORF is not interrupted by any TE either (in fact, there is no TE annotated in zeat2 in Fig 6A, nor on the UCSC Genome Browser repeatmasker track when navigating to the provided coordinates) -- why is zeat2 even a gene of focus for the authors?

- The justification for a distinct ZTA from ZGA is still not adequate. Fig 2A is potentially misleading in this regard (see below) and at odds with Fig 2D, which seems to show the TE genes and non TE genes pretty much span the same range of expression levels. To demonstrate a distinct ZGA, the authors would need to graph *distributions* of expression levels in each gene group at each time point and show statistically significant differences between groups. Regarding Fig 2A -- the top graph appears to show there are overall far fewer TE-alone transcripts than the other categories (and in fact, the overlapping y-axis scales are very misleading in this regard). The bottom graph appears to show summed expression levels across all genes in a group, which have very different numbers of genes. So, the sum will scale with the number of genes included in a group, thus the largest groups will have the highest sum. A bulk-level analysis like this that ignores group sizes

and variation within groups is not adequate to support the authors' claims of a distinct ZTA

Related, Page 9 Line 1 -- citation 24 is a review paper and thus is not the primary source for the timing of minor/major ZGA. At 64 cell, only a single (very unique) locus is activated (mir-430), and minor wave proper is considered to be 128c-512c. (Heyn Cell Rep 2014, Hadzhiev Dev Cel 2023). In concordance with this, a more recent characterization (Bhat Cell Rep 2023) did not identify robust non mir-430 nuclear transcription at 64c (they suggest spurious/stochastic expression occurs prior to this). [none of these papers appear to be cited currently]

Additionally, Page 28 Line 11 - the caveat regarding cytoplasmic polyadenylation needs to be stated much earlier, as a large proportion (all?) of the early increases the authors observe (Fig 2A) are likely due to poly(A) tail lengthening of maternal transcripts rather than de novo transcription. This strongly impacts the interpretation of the dynamics observed. In their response to Reviewer 2, the authors do an analysis with a parallel poly(A)+rRNA deplete RNA-seq dataset to show that TE transcripts are not influenced by poly(A) dynamics, though the issue remains that many other mRNA are affected, and this has an impact on comparisons between the different gene classes (Fig 2). PS -- other parallel poly(A)+rRNA deplete datasets exist that *do* have replicates, e.g. Vejnar et al Genome Res 2019.

- If I'm understanding things correctly, there is something off about the N/C proportion analysis that needs to be clarified. The N/C proportion equation (Fig 4A) would seem to suggest that a strongly cytoplasmic TE would have a very small numerator and a larger denominator, leading to a proportion $\ll 1$. Fig 4B is a bulk-level plot, though a distributional plot (boxplot, violin plot) would be more appropriate to demonstrate the variability within/between groups, as well as the range of values -- we would expect both strongly cytoplasmic and strongly nuclear genes, and the shape of the distributions should reflect this. In Fig 4D, the y-axis scale suggests that all the proportions are above 1 ($\log_2 > 0$) -- is this indeed the case? If a smoothing factor was being applied (e.g., +1?), that would heavily change the shape of the distribution and warrant a reanalysis of statistical significance. I would suggest that the N/C proportion formulation needs to be calibrated if the values overall are skewing > 1 so that we better understand what it is representing.

(Remarks on code availability)

Version 2:

Reviewer comments:

Reviewer #3

(Remarks to the Author)

Thank you to the authors for adequately clarifying the points raised in the previous review. I have no further comments

(Remarks on code availability)

Zygotic activation of transposable elements during zebrafish early embryogenesis

Bo Li, Ting Li, Dingjie Wang, Ying Yang, Puwen Tan, Yunhao Wang,
Yun-Gui Yang, Shunji Jia, Kin Fai Au

Responses to reviewers

Reviewer #1

In this manuscript Li and colleagues discover the TEs that are expressed during early zebrafish development. They perform long-read sequencing at 11 stages of early zebrafish development and analyze the specific TE copies that are expressed either as TE-alone or as part of the gene TE-gene. They find two waves of TE expression after ZGA and this was not corroborated by expression analysis using short read sequencing. They also break down the analysis to TE families and subfamilies. They touch upon TE regulation by analyzing some histone modifications and they also analyze the evolutionary aged of TEs and find that younger TEs are the ones that are mostly expressed. Most of the TE transcripts are nuclear. Finally, they do some experiments with two new loci or two new TE-genes that they identify and show that they are important for embryonic development. In my opinion the analysis of TEs in this manuscript is very thorough and the paper has merit as a resource for the community even though its biological aspects may be more limited.

Response summary: We greatly appreciate your valuable suggestions. With your suggestions, we have performed extensive additional analyses and experiments to improve the quality of the manuscript. Below is a brief summary:

- **Histone mark analysis:** We analyzed two active histone marks (H3K4me3 and H3K27ac) with the public data, and found that their patterns are consistent with our results.
- **TF analysis:** We performed transcription factors (TFs) expression and motif prediction analyses to identify potential TFs involved in TE regulation during early development. Also, we conducted comprehensive literature research about TE regulation and have added the corresponding discussion in the revision.
- **Functional analysis of novel genes:** We performed functional analysis on novel genes that are newly characterized in this study.
- **Biological function study of *zeat1* and *zeat2*:** We further explored the biological functions of two novel genes *zeat1* and *zeat2* by new experiments, including studying the essential role for karyomere fusion of *zeat1* and the regulatory role of mRNA processing of *zeat2*.

Please see the details in the point-by-point responses below.

1. The authors should deepen their investigation on the regulations of TE expression. They mostly work on their repression but what happens to their activation. What is the correlation with active histone marks?

Response: Thanks for this thoughtful suggestion. With an extensive literature search, we found two published data sets (Zhang et al. 2018; Zhu et al. 2019) that focused on profiling the active marks during early embryonic development in zebrafish, which could be helpful for our analysis. With Zhang et al. 2018 study, we first analyzed H3K4me3, an active mark around promoters. We found the H3K4me3 deposition level greatly increased from Oocyte to Dome (Response Figure 1a), consistent with the zygotic TE activation and the ATAC-seq results. The pattern of H3K27ac, an active mark around enhancers was also in line with the pattern of zygotic TE activation (Response Figure 1b). The results based on the data of Zhu et al. 2019 further validated our results orthogonally: the H3K4me3 modification level around TE loci remained as low as Oocyte until the 1k-cell stage when ZTA initiates (Response Figure 1c). Taken together, the results from active histone marks reinforce our previous conclusions on the consistency between ZTA and the corresponding epigenetic regulation.

We have added this result to the revised manuscript (Page 19, Line 21-23 and Page 20, Line1-4).

Revised contents in the main text:

“As previously reported, the active mark H3K4me3 exists in sperm yet is removed in the early embryo for both genes and TEs (Supplementary Fig. 16b and Fig. 5d, respectively). Compared to regular genes, which show elevated levels of H3K4me3 at the 128-cell stage, TE-alone loci have a subtle increase at the 1k-cell stage and a significant rise at the dome stage (Fig. 5d,e and Supplementary Fig. 16b,c), which is consistent with the delayed activation of TEs versus genes. Another active mark H3K27ac also demonstrates a similar regulatory pattern (Fig. 5f and Supplementary Fig. 16d)”

Response Figure 1. Active histone modification marks during ZTA. (a) and (b) H3K4me3 and H3K27ac (Zhang et al. 2018) increase during ZTA. (c) H3K4me3 (Zhu et al. 2019) increases during ZTA until the 1k-cell stage.

2. Are there any specific transcription factors that regulate TE expression or repression? Can the author perform motif analysis?

Response: We performed the literature review and bioinformatics analysis to enhance our understanding of TF regulation on TE activation and repression.

Literature review: Several known transcription factors regulate TE expression and repression in mammals. For example, human cleavage-specific homeodomain transcription factor DUX4 and mouse ortholog DUX can regulate HERVL and MERVL family in human and mouse, respectively (Hendrickson et al. 2017). Klf5 is another transcription factor that can regulate the activation of three ERV families in mouse 2C blastomeres (Kinisu et al. 2021). In addition, Gata2, Rara Zfp281 and Tbx-family factors are also validated to regulate MERVL (Kinisu et al. 2021). However, to the best of our knowledge, transcription factors for TE activation in zebrafish have been rarely studied. The role of *mxtx1/2* revealed by our study would be likely the first TFs that can be hypothesized to regulate ERV activation in zebrafish despite lacking direct experimental evidence. In this revised manuscript, we have added another piece of evidence to support the regulatory role of the *mxtx1/2* on ERV by examining previously published ChIP-seq data (Xu et al. 2012). The results supported the specific binding of *mxtx2* on the ERVs, which indicates *mxtx2* as a transcriptional activator over TE activation (see the details below in Response Figure 10).

Per TE repression, the Kruppel-associated box zinc finger proteins (KRAB-ZFPs) have been revealed to be involved in silencing TE transcription by recruiting the transcription regulator TRIM28 and H3K9me3 mediators (Imbeault et al. 2017; Yang et al. 2017). However, KRAB-ZFPs gene family is absent from fish species, including zebrafish (Imbeault et al. 2017). Recently, the Feschotte lab has reported that a large ZNF gene family can repress the TE expression during the early embryonic development in zebrafish, and highlighted the potential conserved approach in TE silencing (Wells et al. 2023).

Bioinformatics analysis: To predict the potential TFs in regulating TE expression or repression, we first run FIMO function within the MEME Suite to search the known motifs for 550 TE loci and identified 486 TFs with potential binding sites ($P < 0.00001$) on TE's promoter regions (Response Figure 2a). We then investigated the expression pattern of 2,546 annotated TFs (downloaded from AnimalTFDB 4.0 (Shen et al. 2023)) during early embryonic development in zebrafish and found that 2,120 TFs have TPM ≥ 1 . Among 486 potential TE-regulating TFs, 469 are highly expressed during early development in zebrafish, which can be grouped into two major clusters (Cluster1 and Cluster2) by their expression pattern (Response Figure 2b). Interestingly, TFs within Cluster1 are more likely to be expressed before and during ZTA, while those within Cluster2 are more likely to be expressed after ZTA or later stages. This result may lay a foundation for further identifying the roles of TE-related TFs during zebrafish early development, although it is beyond the scope of this manuscript. Therefore, we have added a summary and a discussion of these results in the Discussion section (Page 29, Line 3-9) and a detailed description in Supplementary Note 12 and Supplementary Fig. 22.

Revised contents in the main text:

"A few TFs were reported to be involved in the TE activation (e.g., Dux, Klf5, Gata2) and the TE silencing (e.g., KRAB-ZFPs). A recent study showed that a large ZNF gene family represses TE expression in zebrafish, underscoring a potential conserved approach in TE silencing as mammals. In addition to mxtx1/2, we also analyzed 2,546 annotated zebrafish TFs and found that 469 TFs were highly expressed with motifs identified at the activated TE-alone loci during ZTA (Supplementary Fig. 21 and Supplementary Note 12)."

Response Figure 2. Prediction on transcription factors potentially regulating TE during zebrafish early embryonic development. (a) Pipeline for TF prediction. **(b)** The expression pattern of 469 TFs during early embryonic development. Cluster1 represents TFs that are highly expressed until 75%-epiboly as TEs are highly activated, and Cluster2 contains TFs that are highly expressed later during the repression period of ZTA.

3. Can the authors discuss what is the potential functions of the TEs? What is known about TEs in early zebrafish development but also in the development of different organs?

Response: TEs have been shown to actively influence mammalian-specific developmental processes, especially during pre-implantation, extra-embryonic development, and at the maternal-fetal interface. However, the roles of TEs in zebrafish embryogenesis and organ development remain largely unexplored. To date, only a few TE families have been identified as markers of specific embryonic stages in zebrafish. For example, BHIKHARI, a family of ERVs in zebrafish, is exclusively expressed in the mesendoderm lineage during gastrulation (Vogel and Gerster 1999; Chang et al. 2022). Similarly, another ERV, *crestin* (also known as BHIKHARI-2), serves as a specific marker of the neural crest (Rubinstein et al. 2000; Luo et al. 2001). These previous studies suggest that specific TEs may have distinct roles in zebrafish development, but further research is needed to elucidate their precise functions across different stages and organs.

We have added the following paragraph in the Introduction (Page 3, Line 10-18).

Revised contents in the main text:

“While dysregulated TE expression has been strongly associated with developmental defects, there is growing urgency to achieve a more accurate and comprehensive understanding of TEs and their broader biological impact. The community is making progress towards resolving this problem. For instance, recent studies based on single-cell RNA-seq have indicated that cell type-specific activation of different TEs may contribute to tissue-specific developmental processes. Recently, the activation of a particular endogenous retrovirus has been implicated as critical for mesoderm development in zebrafish.”

4. The authors could describe better the newly found genes.

Response: To better understand the newly found genes in this study, we carried out several downstream analyses with 2,895 transcript sequences from novel gene and TE-gene loci. First, we predicted the coding potential for these transcripts using CPC (<https://cpc.gao-lab.org/>), which resulted in 1,622 protein-coding and 1,273 non-coding RNAs. Second, we conducted functional analysis on the 1,622 protein-coding mRNAs. We extracted the protein sequences of the longest mRNA to represent each gene and used InterProScan for functional annotation (Response Figure 3a). In addition, we aligned protein sequences of all novel genes against all protein sequences of human and zebrafish and identified 367 genes with homologs in both or either species (Response Figure 3b).

We have added this analysis to the Results section of this revised manuscript (Page 22, Line 14-17).

Revised contents in the main text:

“Our comprehensive transcriptome has cataloged 1,674 novel gene loci (including TE-genes and genes) that were previously unannotated by the reference annotation library Ensembl. Among these, 1,260 have protein-coding potential, 698 of which have known functions and/or conserved domains (Supplementary Fig. 18d,e).”

Response Figure 3. Functional annotation on the unannotated genes. (a) Functional annotation pipeline for the novel genes identified in this study. (b) Homolog search for the novel genes.

5. It is not obvious from the text what is new data and what is reanalyzed data

Response: To clearly distinguish between new data and reanalyzed data, we explicitly indicate if the data used in each analysis is publicly available. This has been modified in the revised manuscript (Page 9, Line 14-15; Page 10, Line 6; Page 10, Line 22-23; and Page 19, Line 14-15). We also provided a spreadsheet to describe the public data used in this study (Supplementary Data S13).

6. What is the function of *zeat1* and 2. A brief description is provided but the authors don't dig deeper to the function of these chimeric genes

Response: Our findings in the original manuscript suggest that both *zeat1* and *zeat2* play crucial roles in early zebrafish development. In the revised manuscript, we have delved deeper into the functional characterization of these two novel genes to provide a more comprehensive

understanding of their roles in zebrafish development. We have expanded our analysis to explore their specific contributions to developmental processes.

Biological function of *zeat1*.

To elucidate the function of the *zeat1* gene, we performed microinjections of H1m-GFP protein (a zebrafish H1 type linker histone that labels the chromatin) and wheat germ agglutinin (WGA) Alexa Fluor 647 conjugate (WGA-Alexa 647) into wild-type (WT) and maternal *zeat1* (*Mzeat1*) mutant embryos at the one-cell stage. These injections labeled the nuclei and nuclear envelope, respectively. Time-lapse confocal microscopy revealed that *Mzeat1* mutant embryos exhibited severely fragmented and morphologically abnormal nuclei (Response Figure 4).

Karyomeres are intermediate cleavage-stage structures where individual or groups of chromosomes are enclosed by the nuclear envelope, and they fuse to form a mononucleus (Schoft et al. 2003). High-resolution imaging of individual nuclei at distinct mitotic cell-cycle transition points, particularly from telophase to interphase, demonstrated that the abnormal nuclear morphology observed in *Mzeat1* mutants is due to the karyomere separation from each other. These envelopes fail to fuse to form a mononucleus, resulting in a multimicronucleated phenotype (Response Figure 5). This suggests that Zeat1 is required for karyomere fusion.

Further analysis of Zeat1's subcellular localization showed that the Zeat1-GFP signal colocalized with WGA-Alexa 647, which indicated that Zeat1 localized to the nuclear envelope, consistent with its proposed role in nuclear membrane fusion (Response Figure 6).

Collectively, these findings further revealed that *zeat1* is a novel nuclear envelope-associated protein that is essential for karyomere fusion during early zebrafish embryogenesis.

Response Figure 4. *zeat1* is required for proper nuclear morphology. Interphase nuclei and nuclear envelopes were labeled with H1m-GFP protein and WGA-Alexa 647 in WT and *Mzeat1* mutant at the 64-cell stage.

Response Figure 5. Imaging of individual nuclei at distinct mitotic cell-cycle transition points. Cell cycle time course from the 64-cell to 128-cell transition. Images are projections of multiple confocal Z-slices. For each time point, four embryos were examined. In each case, multiple nuclei or cells of each embryo were also examined.

Response Figure 6. Subcellular localization of Zeat1 protein. WT embryos were co-injected with *zeat1*-GFP mRNA and WGA-Alexa 647. The Zeat1-GFP fusion protein was observed to localize at the nuclear envelope, as indicated by the colocalization of the Zeat1-GFP signal with WGA-Alexa 647.

Biological function of *zeat2*.

Our bioinformatics analysis in the original manuscript suggested that *zeat2* is the zebrafish ortholog of the human *FASTKD5* gene, which is involved in the processing of mitochondrial precursor mRNAs that lack flanking tRNAs (non-canonical RNA processing) (Ohkubo et al. 2021). Based on this, we hypothesized that *zeat2* may play a similar role in regulating mitochondrial gene processing in zebrafish. The zebrafish mitochondrial genome encodes 13 protein-coding genes, some of which have non-canonical junctions, including the 5'UTR of *co1*, the 3'UTR of *nd6*, the region between *atp6* and *co3*, and the region between *nd5* and *cyb*, which are all potential targets of *zeat2* (Broughton et al. 2001; Ohkubo et al. 2021) (Response Figure 7a).

To investigate whether *zeat2* is responsible for processing these non-canonical cleavage sites, we conducted qRT-PCR across the junctions in WT and zygotic *zeat2* (*zeat2*^{-/-}) mutant embryos to quantify the abundance of mitochondrial transcript precursors for these genes. Our results demonstrated a significant accumulation of the non-canonical precursor mRNAs in the absence of *zeat2*, particularly at the 5' UTR of *co1* and the region between *atp6* and *co3* (Response Figure 7b). In contrast, canonical cleavage sites involving tRNA flanking, such as *nd1-nd2* and *nd3-nd4l*, showed no significant accumulation of precursor transcripts in the *zeat2*^{-/-} mutants compared to WT (Response Figure 7b).

In summary, our new results in this revision indicate that *zeat2*, an ortholog of *FASTKD5*, plays a similar role in processing mitochondrial mRNAs at non-canonical junctions. The loss of *zeat2* results in abnormal processing of *co1*, *co3*, and other mitochondrial genes, which in turn probably affects the oxidative phosphorylation system, leading to developmental arrest in zebrafish embryos.

We have added these new results into the Results (Page 23, Line 13-23; Page 24, Line 1-5; Page 24, Line 17-23 and Page 25, Line 1-8).

Revised contents in the main text:

*“To elucidate the function of the *zeat1* gene, we performed microinjections of H1m-GFP protein (a zebrafish H1 type linker histone that labels the chromatin) and wheat germ agglutinin (WGA) Alexa Fluor 647 conjugate (WGA-Alexa 647) into wild-type (WT) and *Mzeat1* mutant embryos at the one-cell stage. These injections labeled the nuclei and nuclear envelope, respectively. Time-lapse confocal microscopy revealed that *Mzeat1* mutant embryos showed severely abnormal nuclear morphology that was characterized by fragmented chromatin enclosed by the nuclear envelope (Fig. 6d).*

*Further investigation of the entire cell cycle suggested that karyomere fusion was disrupted in *Mzeat1* mutants, resulting in multimicronucleated cells during the transition from telophase to interphase (Fig. 6e and Supplementary Movie 1,2). Karyomeres, which are intermediate cleavage-stage structures with individual or groups of chromosomes enclosed by the nuclear envelope, failed to fuse properly, as seen in similar *bmb* maternal mutants⁵¹. The localization of *zeat1* to the nuclear envelope further supports its role in facilitating nuclear envelope fusion (Supplementary Fig. 19e). These results show that *zeat1* is a critical nuclear envelope-associated protein and is essential for karyomere fusion during early zebrafish embryogenesis.”*

*“To determine whether *zeat2* regulates non-canonical mitochondrial RNA processing like *FASTKD5* in humans, we performed qRT-PCR to quantify the abundance of mitochondrial transcript precursors between WT and *zeat2*^{-/-} mutant embryos. The zebrafish mitochondrial genome encodes 13 protein-coding genes, some of which have non-canonical junctions, including the ones in the 5'UTR of *co1*, the 3'UTR of *nd6*, and regions between *atp6* and *co3*, and *nd5* and *cyb* (Supplementary Fig. 19f). We observed a significant accumulation of non-canonical precursor mRNAs in the absence of *zeat2*, with a particularly 27-fold enrichment of the 5' UTR of *co1* (Fig. 6h). In contrast, canonical cleavage sites involving tRNA flanking, such as *nd1-nd2* and *nd3-nd4l*, showed no significant precursor accumulation in *zeat2*^{-/-} mutants compared to WT (Fig. 6h). These results indicate that *zeat2* is an ortholog of *FASTKD5* and plays a similar role in processing mitochondrial mRNAs at non-canonical junctions. The loss of *zeat2* disrupts the processing of *co1*, *co3*, and other mitochondrial genes, which in turn probably affects the oxidative phosphorylation system and leads to developmental arrest in zebrafish embryos.”*

Response Figure 7. Processing of mitochondrial precursor mRNAs in *zeat2* mutants. (a) Schematic representation of the zebrafish mitochondrial genome, with each gene colored and labeled accordingly. Arrows indicate the specific genomic regions where primers were designed for the qRT-PCR assay. **(b)** qRT-PCR results showing the processing of mitochondrial precursor mRNAs in *zeat2*^{-/-} mutants compared to WT. The results of qRT-PCR. 18S rRNA was used as the internal control. An unpaired T-test was applied to determine statistical significance (**** $P < 0.0001$ and * $P < 0.05$)

Reviewer #2

This manuscript reports findings on the activation of transposable element (TE) expression during the early stages of zebrafish embryogenesis. Using long-read sequencing, the authors are able to resolve TE expression at the locus level, and find that only a small subset of TE loci produce transcripts (as opposed to chimeric gene-TE transcripts). They demonstrate that zygotic TE activation (ZTA) follows zygotic gene activation (ZGA), but apparently also proceeds in two waves. I enjoyed reading this manuscript and thought there were several interesting analyses. The discovery of ZTA, temporally distinct from ZGA, is novel and has (to my knowledge) not been reported before, although as the authors note, future studies with long-read sequencing will be needed to determine whether this is specific to zebrafish or a general feature of other animals. This paper will be an important reference for people working in zebrafish embryology, and further advances the use of this organism as a model for studying the activity of TEs in early development. Overall, it is clear that the authors have put a lot of effort into this project, and the experiments are clearly described and appear to have been rigorously executed. I have only one major concern, and am certain that this can be addressed without significantly changing the manuscript, but will possibly some re-writing and cutting of small sections. Other than that, I have several other suggestions to improve the manuscript, most of which can hopefully be addressed without the need for further analyses. Feel free to concentrate on the ones you feel are most important and ignore some others.

Response summary: Thank you for the kind and thoughtful comments. In sum, we have performed the following new analyses and made extensive revisions to the manuscript based on these valuable suggestions:

- **The impact of cytoplasmic polyadenylation.** We re-analyzed a publicly available RNA-seq dataset that covered the zebrafish early development (0-8 hours post fertilization, hpf) generated by both poly(A) capture and ribosomal RNA depletion library preparation methods. These results suggested that the expression pattern of TE-alone transcripts may not be greatly impacted by this prevalent phenomenon.
- **Two-wave of ZTA.** In the revised manuscript, we withdrew the claim of “two waves of ZTA” due to the lack of direct evidence to clarify the difference.
- **Clustering of TE transcripts.** We have added an explanation of the clustering strategy to show that eight clusters are appropriate for our current analysis.
- **Subcellular localization of TEs.** We performed FISH experiments on three selected TE subfamilies—hAT-N76 (DNA transposon), BHIKHARI_I (LTR), and L2-1 (LINE)—at later developmental stages (bud and 6-somite stages). The results revealed a dynamic subcellular localization pattern for the LTR subfamily, BHIKHARI_I, which initially showed nuclear localization at the shield stage, shifted to the cytoplasm at bud and 6-somite stages. This indicates that BHIKHARI_I transcripts eventually move to the cytoplasm but may not be re-imported into the nucleus. These new results enhance our understanding of the dynamic subcellular localization of TE subfamilies.
- **Discussion of several suggested issues.** In the revised manuscript, we updated several parts in the Results and Discussion sections based on the suggestions.

We hope the revised manuscript has solved most of your concerns. Please see the following point-by-point responses.

Major concern:1. An important note about poly-A enrichment methods and maternally deposited transcripts: This is one aspect of the paper that does require significant revision, or at least a more thorough disclaimer for interpreting results. In animals (and certainly zebrafish:<https://doi.org/10.1242/dev.159566>), maternally deposited transcripts undergo cytoplasmic polyadenylation, and thus are initially much harder to detect via poly-A enrichment-based sequencing platforms. The authors do briefly mention polyadenylation in the first paragraph of the discussion, but I think this is a bigger issue than that, as it could lead to dramatic underestimation in the number of maternally deposited transcripts, and skewing of the trajectories described in fig 3d.

Response: We very much appreciate this important reminder and the comments. We realized that cytoplasmic polyadenylation of maternal mRNA could dramatically influence the characterization of expression landscape during zebrafish early embryonic development due to the quantification bias caused by poly(A) enrichment library preparation. Several publications, including the one suggested by the reviewer, underscored that cytoplasmic polyadenylation is quite prevalent for zebrafish maternal mRNAs (Aanes et al. 2011; Winata et al. 2018). Due to the delay of polyadenylation of maternal mRNA, the abundance of a subset of genes could be underestimated by the poly(A) enrichment approach. However, we found that this cytoplasmic polyadenylation would not largely influence the quantification of our research targets, i.e., TE-alone transcripts and its expression pattern as presented in Fig. 3d because TE-alone transcripts account for a tiny proportion of maternal mRNAs.

To further confirm our conclusion, we analyzed publicly available RNA-seq data (PRJNA624126) covering the zebrafish early development (0-8 hpf) generated by poly(A) capture and ribosomal RNA depletion library preparation methods. Theoretically, if one transcript is greatly influenced by cytoplasmic polyadenylation, the abundance estimated by poly(A) capture should be lower than the one by ribosomal RNA depletion due to the delayed polyadenylation.

For regular genes, we identified a considerable number of maternal mRNAs with quantification bias (i.e. abundance underestimation by poly(A) capture method), and the overall trend gradually declined during the onset of early development. However, for TE-alone mRNA, the bias is not significant (Response Figure 8). Therefore, the expression landscape and corresponding trajectories presented in Fig. 3d is not be influenced largely by the poly-A enrichment strategy.

We should be also aware that this public RNA-seq dataset does not have replicates, so a robust conclusion cannot be drawn on which transcripts are affected to what extent. Of note, abundance estimation bias could be also resulted from other factors, such as bias in sequencing. In fact, for each stage that we analyzed, there existed many transcripts with a higher abundance estimated by poly-A enrichment method compared to the one by rRNA depletion. In this revision, we have summarized these results and added the discussion in the Discussion section (Page 28, Line 11-15).

Revised contents in the main text:

“Cytoplasmic polyadenylation prevalently existing in zebrafish early development may also lead to a bias in quantifying maternal TE transcripts albeit significant underestimation is not observed for TE-alone

transcripts between poly(A) enrichment-based and ribosomal RNA depletion-based RNA-seq (Supplementary Fig. 20 and Supplementary Note 11).”

Response Figure 8. Quantification difference between two library preparations (Ribosomal RNA depletion and poly(A) enrichment) for mRNA sequencing. Each dot in the plot represents a transcript identified in this study. x-lab represents the \log_2 transformed TPM quantified by poly(A) capture library preparation, and y-lab represents the \log_2 transformed TPM quantified by ribosomal RNA depletion method. Compared to regular genes, TE-alone transcript abundance estimation has a good correlation between the two methods.

Minor concerns/suggestions:

2. The authors identify 706 TE-alone transcripts deriving from 550 distinct loci. Given the abundance and diversity of young TE families in the zebrafish genome, this is a remarkably small number. It would be useful to get an idea of how conservative this estimate is, and the degree to

which it is influenced by the thresholds used for defining “TE-alone”, or methodological aspects such as sequencing depth. In particular, I think it is important to briefly point out that the approach does not take into account TE insertion polymorphisms. Zebrafish are known to be highly heterozygous, and there are certainly many polymorphic TE loci between individuals. Thus, it is possible (even likely) that some TE-alone transcripts are discarded because they arise from loci that exist in the sequenced individuals, but not the reference genome.

Response: We agree that it is necessary to discuss how the strategy of identifying active TE transcripts in zebrafish influences the results. Overall, we adopted stringent criteria to annotate the active TE-alone loci and the estimate tends to be conservative. This number of identified active TE-alone loci can be affected by the following major factors:

(1) Definition of TE-alone transcripts. Our primary definition of TE-alone transcripts requires both the 5’end of a transcript within a TE region (TE-promoter driven transcription) and over 90% of the transcript sequences are TE. If we reduced this threshold to 70%, 1818 TE-alone transcripts and 820 TE-alone loci would be selected, compared to the correspond numbers 1714 and 762, respectively. Thus, the influence is not critical. This primary screening was subsequently followed by manual annotation to further remove errors.

(2) Stage-specific expression. We noticed that a considerable number of TEs are expressed in specific stages. In our study, although we have included 11 different stages, it is very likely that we may miss the TE-alone transcripts transiently expressed at other developmental time points, especially after the shield stages.

(3) Sequencing depth. Despite the high sequencing quality for our dataset, the average total reads for each stage is ~1 million. Our previous long-read-based study of human transcriptome has shown this read number could provide good coverage of transcripts with reasonable expression (Au et al. 2013; Weirather et al. 2017), yet increasing sequencing depth can improve the discovery of the lowly expressed TE-alone transcripts.

(4) TE polymorphism. The TU zebrafish genome may not include all active TEs in our sequenced samples, which indeed causes a lower estimate of TE number. To roughly estimate the active TE-alone loci across the zebrafish genome, we aligned all 550 TE-alone loci identified in this study onto the reference genome (downloaded from Ensembl) and identified 756 additional genomic regions with high similarity, which may have expression potential at other developmental stages or various contexts.

According to these analyses, we have added the following points to the Discussion section (Page 27, Line 21-26 and Page 28, Line 1-2):

Revised contents in the main text:

“It is worth noting that 550 TE-alone loci is a conservative estimate and several factors may influence the identification of active TE-alone loci, including stage-specific TE expression, low sequencing coverage, TE insertion polymorphisms among zebrafish strains and the TE annotation methods. Additionally, TE heterogeneity among zebrafish strains may contribute to a biased estimate of TE-alone loci. The TE polymorphism between the experimental samples (AB and India strains) and the TU reference genome may lead to the misidentification of highly divergent TE loci and strain-specific copies.”

3. The authors use the term “active” to describe TEs throughout the manuscript. It would be good to briefly clarify somewhere in the introduction or first results paragraph that you are referring to transcriptional activity, not necessarily transpositional activity.

Response: We have added a sentence in the Introduction to emphasize the definition of “active” in the manuscript (Page 4, Line 19-21)

Revised contents in the main text:

“Of note, in this study, active TE is defined by transcriptional activity but not necessarily transpositional activity.”

4. I thought the analysis of core domain losses in TE-alone transcripts was interesting. While this does suggest that many families are incapable of autonomous transposition, I think it would be good to emphasize that they are transposing nonetheless! This is clear from the fact that several TE families have many full-length, nearly identical copies in the genome, despite lacking many of the core domains required for fully autonomous transposition. For example, BHIKHARI encodes only a single Gag protein, and yet is very likely still actively transposing in zebrafish populations (doi:10.1101/2024.03.25.586437v1).

Response: We totally agree with this argument. We have added the following discussion to the Discussion in the revised manuscript (Page 30, Line 15-18). And also cited the paper suggested (Page 30, Line 18).

Revised contents in the main text:

“This result suggests that some TEs can still be transposed without intact structure of domains. For example, an endogenous retrovirus in zebrafish, BHIKHARI is likely actively transposed despite that it only encodes a Gag protein.”

5. With respect to the timing of ZTA, I agree with the authors’ suggestion that it likely relies on host factors provided by ZGA, thus explaining the relative delay of ZTA. However, I am more skeptical about the existence of two distinct ZTA waves. Do the authors reach this conclusion solely based on the correlation analysis shown in Fig 2e? Are there consistent differences between the TEs activated at 1k versus oblong stage? E.g. retroelements in the former, DNA transposons in the latter? Is there any evidence to suggest that major and minor waves of TE activation would be conserved in other species is this just a quirk of the TE composition in zebrafish.

Response: Thank you for the suggestions. We agree that more evidence is needed to support the claim of two waves in the activation pattern of zygotic transposable elements during early zebrafish development. Therefore, the concept of “two waves of ZTA” has been withdrawn in this revision.

6. The authors perform a detailed analysis of TE-alone transcript trajectories in Fig. 3. I liked this section but would be cautious about over-interpreting the differences between the categories described in Fig. 3D. Many of these patterns look qualitatively similar to one another, if allowing for some noise and variation. Focusing on BHIKHARI again, the authors report that the transcripts

can be separated into six different trajectory types. However, as the authors note, the regulatory regions of BHIKHARI are highly similar to one another, and have clearly conserved regulatory elements. It would therefore be surprising if the expression patterns really did fall into distinct groups, as opposed to being an artifact of the threshold chosen for the hierarchical clustering step. From looking at the dendrogram in Fig 3D, it looks like 3 or 4 clusters might be a more natural cut point, and it would be interesting to see if this modification would lead to more BHIKHARI transcripts falling within the same cluster. The authors attribute the intra-family heterogeneity in trajectory types to the “possible existence of unknown regulatory sequence variations or epigenetics layers”, and I agree with this, but think it might be overstating the case – we already know that different TEs (particularly ERVs/LTRs) recruit distinct sets of transcription factors, and as for epigenetic layers, since TEs insert (mostly) randomly into the genome, it is natural that there would be variation in transcriptional strength and pattern due to the varying genomic contexts that they find themselves in.

Response: Thanks for these comments and suggestions. Indeed, we share similar thoughts on the possible roles of the varying genomic contexts of each TE copy. Below we discuss the choices of cluster number and the possible interpretation in epigenetics in more detail:

The optimal number of clusters in Fig.3d.

We agree that finding the optimal number of clusters for the heatmap in Fig. 3d could be challenging, as it is open to interpretation. In this study, we chose 8 clusters of expression trajectories for two specific reasons: (1) the expression pattern (trends in expression levels across different developmental stages); (2) the activation and peak timing (the stage at which TE-alone transcripts become detectable).

We found that the choice of 8 clusters could distinguish these subtle but meaningful differences better than a small number of clusters. For instance, while Cluster 2 and Cluster 3 share similar expression patterns (Fig. 3d), they are distinguished by their activation times: Cluster 2 primarily captures transcripts activated at the shield stage, whereas Cluster 3 mainly contains transcripts activated earlier and reaches the peak at the shield stage. Similarly, Clusters 6, 7, and 8 show similar expression patterns, but their peak stages differ (Cluster 6 reaches the peak at the 50%-epiboly stage, Cluster 7 reaches the peak at the dome stage and maintains certain expression levels until the shield stage, and Cluster 8 reaches the peak at the 30%-epiboly stage). These subtle differences could be informative clues for further investigating the distinct regulatory mechanisms of TE-alone transcripts. When we tested 4 as the number of clusters: current Clusters 1, 2, and 3 were combined; Clusters 4 and 5 were combined; Clusters 7 and 8 were combined, where the abovementioned differences were overlooked.

Since we considered 11 stages here, there could exist $2^{11}=2048$ possible binary-based clusters. Therefore, 8 clusters with sufficient members are reasonable to capture meaningful expression differences rather than random noise. In fact, the smallest cluster (Cluster 4) contains 35 transcripts (Fig. 3d) --- it is not very likely to have this number of transcripts to share the same expression and activation pattern by chance over 11 stages.

Overstating the epigenetic regulation.

We agree that our previous description “possible existence of unknown regulatory sequence variations or epigenetics layers” is not appropriate due to the reasons you mentioned here. Therefore, in the revision, we have changed the “unknown regulatory sequence variations” to the “*different epigenetics layers of regulation and the varying genomic contexts*” (Page 14, Line 1).

7. I liked the analysis of subcellular localization differences, but think that the interpretation could be improved. The finding that DNA transposons tend to have lower N/C ratios than LTRs/LINEs is consistent with known mechanisms of replication – since DNA transposons do not use an mRNA template to replicate, they have no need to transport transcripts to the nucleus. LINEs replicate through target-primed reverse transcription, and thus explicitly need to transport their mRNA template to the nucleus, and should thus have high N/C ratios. However, it is somewhat surprising that LTRs have high N/C ratios, since reverse transcription takes place in the Gag capsid in the cytoplasm (as far as we currently understand), and the mRNA template is degraded in the process. Thus, finding high levels of LTR mRNA in the nucleus might suggest incomplete reverse transcription and unintended import of mRNA rather than cDNA. Incidentally, the fact that LTRs and LINEs are enriched in the nucleus at all is strong circumstantial evidence for active transposition in zebrafish!

Response: We very much appreciated your insightful explanation regarding how different mechanisms of transposition among DNA transposons, LTRs, and LINEs could influence the subcellular localization of TEs. Your point about re-importation to the nucleus after translation adds an important layer to understanding the N/C ratio, which we have now incorporated into the revised manuscript to explain these differences comprehensively.

Additionally, we observed that the N/C ratio is quite dynamic for some TEs. For example, performing FISH experiments on three selected TE subfamilies—hAT-N76 (DNA transposon), BHIKHARI_I (LTR), and L2-1 (LINE)—at the later developmental stages (bud and 6-somite stages), we noted distinct patterns. For instance, BHIKHARI_I showed clear cytoplasmic localization at the bud stage, and it was almost entirely localized in the cytoplasm by the 6-somite stage. This may suggest that nearly all BHIKHARI_I transcripts eventually move to the cytoplasm but may not undergo re-importation into the nucleus, as you suggested (Response Figure 10). Therefore, in this revision, we emphasized that the N/C ratio captured in our analysis represents only a snapshot of the entire developmental process, acknowledging that these ratios may change dynamically over time.

We have added a paragraph to the Results to describe our new results and your suggested explanation (Page 17, Line 5-21).

Revised contents in the main text:

“To illustrate these observations, we applied fluorescence in situ hybridization (FISH) to investigate the dynamic subcellular localization of several TE subfamilies. Our analysis reveals that the DNA transposon subfamily hAT-N76_DR predominantly localizes in the cytoplasm, while the LINE subfamily L2-1_DR is localized in the nucleus, consistently from the shield stage to the 6-somite stages (Fig. 4e, f and Supplementary Fig. 15b, c). These findings align with a previous study in mouse embryonic stem cells, which demonstrated a nuclear bias of LINE1 transcripts, suggesting potential regulatory roles in early development. This parallel implies a similar function for LINE transcripts in zebrafish embryogenesis. The transposition mechanisms likely play a key role in influencing the subcellular localization of TEs. For instance, LINEs replicate via target-primed reverse transcription, requiring their mRNA template to be transported back into the nucleus. This process may likely contribute to the observed N/C proportion. In addition, the LTR subfamily BHIKHARI_I displays a more dynamic subcellular distribution across developmental stages. It shows a nuclear bias at the shield stage but shifts to clear cytoplasmic localization at the bud and 6-somite stages (Fig. 4g and Supplementary Fig. 15d). This dynamic localization pattern aligns with above speculations (Fig. 4d), highlighting a more flexible subcellular positioning for LTRs during development.”

8. Since the authors discuss H3K9me3 and piRNAs, they might be interested to know that a large family of zinc fingers proteins (FiNZ-zinc fingers) have recently been shown to be repressing TE activity in zebrafish, likely through recruiting H3K9me3 to TE loci, as is the case for KRAB-zinc fingers in mammals (doi: 10.1101/gr.277966.123).

Response: Thanks for sharing this important paper with us! We have added a paragraph in the Discussion to describe the up-to-date knowledge of TFs on regulating TEs (Page 29, Line 3-9) and cited this paper (Page 29, Line 6).

Revised contents in the main text:

“A few TFs were previously reported to be involved in the TE activation (e.g., Dux, Klf5, Gata2) and the TE silencing (e.g., KRAB-ZFPs). A recent study showed that a large ZNF gene family represses TE expression in zebrafish, underscoring a potential conserved approach in TE silencing as mammals. In addition to mxtx1/2, we also analyzed 2,546 annotated zebrafish TFs and found that 469 TFs were highly expressed with motifs identified at the activated TE-alone loci during ZTA (Supplementary Fig. 21 and Supplementary Note 12).”

9. That being said, I couldn't find scripts or notebooks used for generating figures, which is a shame since they are very attractive! It would be helpful to make these available in the github repo too.

Response: We are glad that the scripts could be helpful. We have uploaded an R markdown file for figure generation used in this study to Github (<https://github.com/Augroup/aTEA>).

Reviewer #3

Li, Li, Wang et al present a detailed analysis of transposable element (TE) expression during the zebrafish maternal-to-zygotic transition. Their use of long-read RNA-sequencing along with conventional short-read RNA-seq expands in a tight time course reveals insights into the temporal dynamics of TE activation, thus expanding on the previous work of Chang et al 2022 that originally mapped TE expression across zebrafish development. For this reason, this paper is a solid contribution to the field. However, the manuscript as written overreaches somewhat in the novelty and significance of their findings, especially regarding the functional consequences of TE expression in the early embryo, and in revision, care should be taken to frame these results appropriately.

Response summary: Thank you for the kind comments and suggestions. With your suggestions, we mainly complete the following experiments and manuscript revisions to improve the rigor and quality of this study.

- **Appropriate interpretation of the results.** Following the suggestions, we carefully have tailored the manuscript to make appropriate descriptions of the novelty of the findings.
- **Removal of “two waves of ZTA”.** Due to the lack of direct evidence, we have withdrawn this claim in the revised manuscript.
- **Novel gene functions.** We further performed several experiments to enhance our understanding of the biological functions of *zeat1* and *zeat2*.
- **FISH experiment at later developmental stages.** We conducted FISH validation for three featured TE subfamilies at the bud and 6-somite stages. We found the DNA transposon and LINE subfamilies maintain the subcellular localization pattern as in the shield stage, while the ERV subfamily, BHIKHARI shows cytoplasm localization in the later stage.
- **Two transcription factors, *mxtx1/2*.** In the revision, we have added more information about how to identify the orthologous relationship between them with *DUX*. We also re-analyzed the ChIP-seq data of *Mxtx2* and *Nanog*, and found that *Mxtx2* can specifically bind on ERVs.

The point-by-point responses below include the details.

Major points:

- The expression patterns as exemplified in Fig 2a lead the authors to define and coin the term "Zygotic TE activation" (ZTA) along with distinguishing a minor and major wave of ZTA. It is difficult to see how TE activation patterns differ sufficiently from activation of other lowly expressed genes to warrant defining a "ZTA." Similarly, the inflection point in expression increase that the authors use to distinguish the minor from major ZTA does not convincingly implicate two waves. Different genes have different activation kinetics, even within a wave, owing to their unique regulatory contexts. There is just not enough evidence that TEs as a group have a unique regulatory profile. The authors use of α -Amanitin, pluripotency factor LOF, and cycloheximide RNA-seq data to justify the existence of ZTA all just seem to show that expressed TEs behave like other genes during ZGA.

Response: We appreciate the suggestions regarding the claim of "two distinct ZTA waves". Indeed, despite the results of expression correlations, it requires more extensive evidence and in-depth investigation to confirm this claim. Therefore, based on your suggestion, we have withdrawn this conclusion from the revised manuscript.

The *mxtx1/2* analysis in Fig 2g / Supp Fig 9 in particular seems to suggest that a lot of ERV expression is *not* downstream *mxtx1/2*. By contrast, the minor and major waves of zebrafish zygotic genome activation have strong mechanistic foundations (see, e.g., Hadzhiev et al, Dev Cell 2023). The concept of a distinct ZTA is as yet speculation, and thus should be limited to the discussion.

Response: We agree that not all ERVs are regulated downstream by *mxtx1/2*, although we still found the expression of many ERVs in Fig. 2g and Fig. S9d, which could indeed be influenced by the imperfect blockage in the CHX treatment (Fig. S9c). While CHX treatment significantly reduced *mxtx1/2* expression (along with other transcription factors), there was still detectable expression that may likely result in a lower level of ERV activation than anticipated. We also appreciate your suggestions regarding the *mxtx2* ChIP-seq data, which provides additional support to the specific regulation of ERVs by *mxtx2* (see details below and Response Figure 10).

- It is unclear how the CRISPR LOF analyses demonstrate the functional significance of TEs per se. The TC1 element in "zeat1" seems to be occurring in an extended 3'UTR according to the RefSeq track on the UCSC genome browser (so, not really a chimera, and misleading figure Fig 6a). Given that a lesion in the CDS was generated, the phenotype is almost certainly not due to loss of the TE but rather loss of the encoded protein. Regardless, demonstrating an injected mRNA rescue would strengthen the phenotypic claim.

Response: We apologize for the confusion of the definition of chimera. Throughout the manuscript, we define chimera as "chimeric transcript" instead of "chimeric protein". In the **Supplementary Note 10**, we have mentioned that this TE inserts into the 3'UTR and it does not change the protein-coding sequence. To avoid potential confusion, in this revision, we have added a note "without interrupting the coding sequence" (Page 23, Line 3) to indicate that this TE insertion does not impact the protein-coding region of this transcript.

In addition, we made multiple attempts to rescue the maternal *zeat1* mutant phenotype by injecting *zeat1* mRNA at the 1-cell stage. Unfortunately, these efforts did not correct the early embryonic developmental defects. Despite the unsuccessful rescue attempts, we conducted additional in-depth experiments to investigate the biological functions of *zeat1* (Response Figure 4-7). These new results revealed that Zeat1 localizes to the nuclear envelope and plays a crucial role in karyomere fusion during the mitotic cell cycle. The failure of mRNA rescue may be attributed to the timing of maternal Zeat1 protein function, which begins with the first cell division immediately after fertilization. The time required to synthesize Zeat1 protein from the injected mRNAs may be so long that the rescue attempts at the 1-cell stage were ineffective. Therefore, it is not surprising that *zeat1* mRNA injection was unable to rescue the maternal mutant defects, as this outcome is quite common in such experiments. We believe the results of our other experiments have significantly improved our understanding of *zeat1* and its role in early embryonic development.

For "zeat2", Fig 6b shows a 932bp region, which is at odds with the protein size suggested in Supp Fig 19b; it has no annotated refseq or ensembl gene model, nor does there appear to be

any obvious RNA-seq expression evidence. So, further clarification is needed for what the *zeat2* gene is.

Response: We sincerely apologize for the typos in the genomic coordinates (27233820 and 27232889 should be corrected as 2723820 and 2732889, respectively) in Fig. 6a. We have provided the definition of novel genes in the supplementary materials: in brief, we checked the reference annotation library (e.g., Ensembl) to find whether it is annotated or not.

- The similarities in ERV activation between zebrafish and mammals is absolutely notable; however, the direct analogy to DUX4/Dux, which have roles in ZGA and downstream TE activation, is a big stretch. How the authors concluded that the zebrafish ERVs are orthologs to DUX4/Dux needs to be clarified, since DUX4 and Dux are not even unambiguously orthologs of each other (they have been called "functional orthologs" despite almost completely dissimilar sequence, suggesting that they actually have convergent rather than conserved functions). The authors should revisit how they discuss these observations.

Response: Thank you for the suggestion. The ortholog relationship between *mxtx1/2* and *DUX4* was established by OrthoFinder and PANTHER as the reciprocal best alignment (see <https://www.alliancegenome.org/gene/ZFIN:ZDB-GENE-000710-7#orthology>). In addition, we directly compared the protein sequences of *mxtx1/2* and *DUX4/Dux*, and found low sequence similarity as you mentioned. Therefore, we agree with you that they may be "functional orthologs". Despite the low similarity, these genes are among the best homologs from the all-to-all alignments and therefore classified as orthologs.

Furthermore, based on your suggestion, we re-analyzed the *Mxtx2* ChIP-seq data and found this TF specifically binds to the ERVs, which may support the relationship of functional orthologs between *mxtx1/2* and *DUX4/Dux* (see details below and Response Figure 10).

- It's unclear to what extent the nuclear localization conclusions may be influenced by signal from nascent transcripts. Are TE-associated transcripts longer on average (and thus, slower transcribed)? Do the FISH images look similarly nuclear at later stages? Why is a bulk all-gene "relative N/C ratio" being calculated as opposed to a nuclear/total value for each gene -- could it be that with so many non TE-genes, any compartment bias is just averaged away?

Response:

- **Factors to influence the subcellular localization.** We agree that quite a few factors can influence the nuclear localization of TEs, including the proposed slower transcription, slower exportation into the cytoplasm, and the rate of transportation back into the nucleus (suggested by another reviewer). Based on your kind suggestions, we checked the length distributions for regular genes, TE-gene transcripts and TE-alone transcripts (Response Figure 9) and found the longer length of TE-associated transcripts you proposed. We have added this result in supplementary materials (Supplementary Fig. 15f). However, due to the lack of direct evidence of the length and transcription speed in influencing subcellular localization of TE-associate transcripts, we do not highlight it as a major conclusion in the main text but have added a brief discussion in the Discussion section (Page 29, Line 20-23 and Page 30, Line 1-2).

Revised contents in the main text:

“TE-alone and TE-gene transcripts are more likely localized inside the nuclei based on both RNA-seq analysis and FISH experiments (Fig. 4). Although those TE-associated transcripts are significantly longer than regular gene transcripts (Supplementary Fig. 15e), which may potentially cause a longer time of transcription, the influences of the nascent TE-associated transcripts on the detection of subcellular localization remains unclear and requires further study.”

Response Figure 9. The length distribution for regular genes, TE-gene transcripts and TE-alone transcripts.

- FISH at later stages.** To better capture the dynamics of subcellular localization of different TE subfamilies, we performed FISH experiments on three selected TE subfamilies at two developmental stages following the shield stage: the bud and the 6-somite stages. Interestingly, we observed distinct subcellular localization patterns for BHIKHARI_I subfamilies at later stages, showing clear cytoplasmic localization from the bud stage. This indicates that the previously observed nuclear localization may be due to delayed export into the cytoplasm. In contrast, hAT-N76_DR and L2-1_DR showed stable localization in the cytoplasm and nucleus, respectively (Response Figure 10). These results indicate that subcellular localization is quite dynamic for some TEs, underscoring the need for more extensive assays to fully understand these patterns.

We have added this new result in the Results section (Page 17, Line 5-21).

Revised contents in the main text:

“To illustrate these observations, we applied fluorescence in situ hybridization (FISH) to investigate the dynamic subcellular localization of several TE subfamilies. Our analysis reveals that the DNA transposon subfamily hAT-N76_DR predominantly localizes in the cytoplasm, while the LINE subfamily L2-1_DR is localized in the nucleus, consistently from the shield stage to the 6-somite stages (Fig. 4e, f and Supplementary Fig. 15b, c). These findings align with a previous study in mouse embryonic stem cells, which demonstrated a nuclear bias of LINE1 transcripts, suggesting potential regulatory roles in early development. This parallel implies a similar function for LINE transcripts in zebrafish embryogenesis. The transposition mechanisms likely play a key role in influencing the subcellular localization of TEs. For instance, LINEs replicate via target-primed reverse transcription, requiring their mRNA template to be transported back into the nucleus. This process may likely contribute to the observed N/C proportion. In addition, the LTR subfamily BHIKHARI_I displays a more dynamic subcellular distribution across developmental stages. It shows a nuclear bias at the shield stage but shifts to clear cytoplasmic localization at the bud and 6-somite stages (Fig. 4g and Supplementary Fig. 15d). This dynamic localization pattern aligns with

above speculations (Fig. 4d), highlighting a more flexible subcellular positioning for LTRs during development.”

Response Figure 10. FISH results of three selected TE subfamilies at the bud and 6-somite stages. Green, β -catenin staining for cell membrane; Blue, DAPI staining for nucleus; Red, specific probes against indicated TE-alone transcripts.

- Relative N/C ratio calculation. We used “the relative N/C ratio” for gene, TE-gene, and TE-alone transcripts because in the study we are interested in the overall pattern of subcellular localization of these three groups. This calculation could diminish the influence of varying magnitudes of absolute expression among different transcripts but represents the preference of subcellular localization for different transcripts. Therefore, this value could be comparable among transcripts.

- Concordance (or lack thereof) with the Chang et al Genome Res 2022 study should be made more explicit. E.g., the Chang study finds many more maternally provided TE associated transcripts -- is the discrepancy due to how TEs are defined?

Response: The Chang study adopted short-read sequencing and our study relied on long-read sequencing, which is the key reason of the difference of TE transcript characterization. Extensive studies have shown that long-read sequencing can very much improve the full-length transcript characterization (Au et al. 2013; Weirather et al. 2017; Nudelman et al. 2018; Kovaka et al. 2019). Long-read sequencing is particularly useful to reliably characterize TE transcripts that contain many repetitive sequences. In order to discuss this discrepancy, in the supplementary materials, we included a systematic comparison of the identification of TE-alone transcripts using long-read vs. short-read sequencing, and we showed how the choice of method could significantly influence the number of detected TE-alone transcripts, including the maternal ones (Supplementary Fig. 4). Furthermore, we agree that the different "TE transcript" definitions in two studies affected the identification results. Our definition relied on the full-length transcripts (identified by long reads) plus RepeatMasker annotation followed by manual curation, which should guarantee a high quality of TE transcript characterization.

- The methods suggest that both AB and a wild strain India (is this WIK?) were used to generate the transcriptomics, which we might expect could have divergent TE expression patterns. However, it does not seem that any of the paper's analyses take this into account or acknowledge the possible TE heterogeneity in the collected data (this would seem to complicate the analysis in Supp Fig 14, e.g.). The authors should clarify.

Response: We appreciate these questions and feel sorry for the confusion.

- The India strain is not WIK but another wild zebrafish strain. We acknowledge that TE heterogeneity exists in our transcriptome data and should be discussed in our manuscript to clarify the issue. Therefore, in the revised manuscript, we have added the following paragraph in the Discussion (Page 27, Line 24-26 and Page 28, Line 1-2):

Revised contents in the main text:

“Additionally, TE heterogeneity among zebrafish strains may contribute to a biased estimate of TE-alone loci. The TE polymorphism between the experimental samples (AB and India strains) and the TU reference genome may lead to the misidentification of highly divergent TE loci and strain-specific copies.”

- For the analysis of Supplementary Fig. 14, two key factors can influence the accuracy of the results: the precision of the 550 TE-alone loci and the quality of the genome assemblies. Despite TE heterogeneity, these TE-alone loci should be accurate based on multi-omics evidence. Long-read RNA-seq alignments strongly support that these TE loci are active in the TU genome, even though their genomic locations may vary between AB and India strains. Moreover, the epigenetic landscape, characterized by our TE annotation, further corroborates this expression pattern.

Regarding genome assembly, poor assembly of TE regions could significantly influence comparative genomic analyses, such as misassembly errors being mistaken for TE absence. However, despite this potential issue, the overall conclusion from this analysis is consistent with other findings and prior knowledge: for example, the most expressed TEs are evolutionarily young. Therefore, we believe the overall results and main conclusions of this analysis are reliable.

Additional points:

- "NC ratio" has a very specific usage to refer to volumes, not RNA quantities, especially in the context of genome activation. The authors should find an alternate way to name this value (or, simply use nuclear proportion instead)

Response: We noticed that "NC ratio" is commonly used in ZGA research with a specific meaning. Therefore, in the revision, we have replaced the "NC ratio" with "NC proportion" and added a clear definition of "NC proportion" as the relative mRNA abundance between the nucleus and cytoplasm at the beginning of this section (Page 16, Line 13-14) to avoid confusions.

Revised contents in the main text:

"relative mRNA abundance between the nucleus and cytoplasm (N/C proportion)".

- Fig 4a, the TN and TC variables seem to be used both for TEs and for genes

Response: We sincerely apologize for this error and have corrected it in the revised manuscript.

- Xu et al, Dev Cell 2012 indeed has *mxtx2* ChIP-seq data, which could be useful for the authors (cf Methods section 13)

Response: Thanks for sharing this valuable information! We downloaded the ChIP-seq data for *Mxtx2* and *Nanog*, and performed genome-wide peak calling for these two transcription factors. Using MACS2 with default parameters (see Methods in the revised manuscript), 3,534 and 33,111 peaks were identified across the zebrafish genome for *Mxtx2* and *Nanog*, respectively. We then intersected these peaks with our TE annotation and found 43 *Mxtx2* and 164 *Nanog* peaks overlapped with TE-alone loci. Interestingly, 40 of 43 *Mxtx2* TE-related binding sites are ERVs (the other 3 are DNA transposons) (Response Figure 11a), which is consistent with our previous motif prediction indicating the *Mxtx1/2* binding on ERVs. However, *Nanog* can bind more diverse types of TEs, including 126 LTRs, 24 DNA transposons and 14 LINEs (Response Figure 11b). This result provides a clue to understand whether *mxtx1/2* can regulate the activation of ERVs. More interestingly, this analysis also showed that *Nanog* is a general TF, which has a broader binding on various types of TEs, but *Mxtx2* is a specific TF for ERVs.

We have added these new results to the Results (Page 10, Line 22-23 and Page 11, Line 1-2).

Revised contents in the main text:

“Interestingly, re-analysis of the publicly available ChIP-seq data for zebrafish Nanog and Mxtx232 suggests that Nanog may function as a general TF for various types of TEs, while Mxtx2 may specifically regulate ERVs (Supplementary Fig. 9e,f).”

Response Figure 11. Identification of Mxtx2 and Nanog binding sites onto active TE-alone loci by ChIP-seq data. (a) Mxtx2 peaks were distributed among different TE families. (b) Nanog peaks were distributed among different TE families.

- Probe sequences for the in situ experiments should be provided

Response: Thank you for the reminder. We have provided the information about the probes to Supplementary Data S15.

References

- Aanes H, Winata CL, Lin CH, Chen JP, Srinivasan KG, Lee SG, Lim AY, Hajan HS, Collas P, Bourque G et al. 2011. Zebrafish mRNA sequencing deciphers novelties in transcriptome dynamics during maternal to zygotic transition. *Genome Res* **21**: 1328-1338.
- Au KF, Sebastiano V, Afshar PT, Durruthy JD, Lee L, Williams BA, van Bakel H, Schadt EE, Reijo-Pera RA, Underwood JG et al. 2013. Characterization of the human ESC transcriptome by hybrid sequencing. *Proc Natl Acad Sci U S A* **110**: E4821-4830.
- Broughton RE, Milam JE, Roe BA. 2001. The complete sequence of the zebrafish (*Danio rerio*) mitochondrial genome and evolutionary patterns in vertebrate mitochondrial DNA. *Genome Res* **11**: 1958-1967.
- Chang NC, Rovira Q, Wells J, Feschotte C, Vaquerizas JM. 2022. Zebrafish transposable elements show extensive diversification in age, genomic distribution, and developmental expression. *Genome Res* **32**: 1408-1423.
- Hendrickson PG, Doráis JA, Grow EJ, Whiddon JL, Lim JW, Wike CL, Weaver BD, Pflueger C, Emery BR, Wilcox AL et al. 2017. Conserved roles of mouse DUX and human DUX4 in activating cleavage-stage genes and MERVL/HERVL retrotransposons. *Nat Genet* **49**: 925-934.
- Imbeault M, Helleboid PY, Trono D. 2017. KRAB zinc-finger proteins contribute to the evolution of gene regulatory networks. *Nature* **543**: 550-554.
- Kinisu M, Choi YJ, Cattoglio C, Liu K, Roux de Bezieux H, Valbuena R, Pum N, Dudoit S, Huang H, Xuan Z et al. 2021. Klf5 establishes bi-potential cell fate by dual regulation of ICM and TE specification genes. *Cell Rep* **37**: 109982.
- Kovaka S, Zimin AV, Pertea GM, Razaghi R, Salzberg SL, Pertea M. 2019. Transcriptome assembly from long-read RNA-seq alignments with StringTie2. *Genome Biol* **20**: 278.
- Luo R, An M, Arduini BL, Henion PD. 2001. Specific pan-neural crest expression of zebrafish Crestin throughout embryonic development. *Dev Dyn* **220**: 169-174.
- Nudelman G, Frasca A, Kent B, Sadler KC, Sealfon SC, Walsh MJ, Zaslavsky E. 2018. High resolution annotation of zebrafish transcriptome using long-read sequencing. *Genome Res* **28**: 1415-1425.
- Ohkubo A, Van Haute L, Rudler DL, Stentenbach M, Steiner FA, Rackham O, Minczuk M, Filipovska A, Martinou JC. 2021. The FASTK family proteins fine-tune mitochondrial RNA processing. *PLoS Genet* **17**: e1009873.
- Rubinstein AL, Lee D, Luo R, Henion PD, Halpern ME. 2000. Genes dependent on zebrafish cyclops function identified by AFLP differential gene expression screen. *Genesis* **26**: 86-97.
- Schoft VK, Beauvais AJ, Lang C, Gajewski A, Prüfert K, Winkler C, Akimenko MA, Paulin-Levasseur M, Krohne G. 2003. The lamina-associated polypeptide 2 (LAP2) isoforms beta, gamma and omega of zebrafish: developmental expression and behavior during the cell cycle. *J Cell Sci* **116**: 2505-2517.
- Shen WK, Chen SY, Gan ZQ, Zhang YZ, Yue T, Chen MM, Xue Y, Hu H, Guo AY. 2023. AnimalTFDB 4.0: a comprehensive animal transcription factor database updated with variation and expression annotations. *Nucleic Acids Res* **51**: D39-D45.
- Vogel AM, Gerster T. 1999. Promoter activity of the zebrafish bhikhari retroelement requires an intact activin signaling pathway. *Mech Dev* **85**: 133-146.
- Weirather JL, de Cesare M, Wang Y, Piazza P, Sebastiano V, Wang XJ, Buck D, Au KF. 2017. Comprehensive comparison of Pacific Biosciences and Oxford Nanopore Technologies and their applications to transcriptome analysis. *F1000Res* **6**: 100.
- Wells JN, Chang NC, McCormick J, Coleman C, Ramos N, Jin B, Feschotte C. 2023. Transposable elements drive the evolution of metazoan zinc finger genes. *Genome Res* **33**: 1325-1339.
- Winata CL, Łapiński M, Prysycz L, Vaz C, Bin Ismail MH, Nama S, Hajan HS, Lee SGP, Korzh V, Sampath P et al. 2018. Cytoplasmic polyadenylation-mediated translational control of maternal mRNAs directs maternal-to-zygotic transition. *Development* **145**.
- Xu C, Fan ZP, Müller P, Fogley R, DiBiase A, Trompouki E, Unternaehrer J, Xiong F, Torregroza I, Evans T et al. 2012. Nanog-like regulates endoderm formation through the Mxtx2-Nodal pathway. *Dev Cell* **22**: 625-638.
- Yang P, Wang Y, Macfarlan TS. 2017. The Role of KRAB-ZFPs in Transposable Element Repression and Mammalian Evolution. *Trends Genet* **33**: 871-881.
- Zhang B, Wu X, Zhang W, Shen W, Sun Q, Liu K, Zhang Y, Wang Q, Li Y, Meng A et al. 2018. Widespread Enhancer Dememorization and Promoter Priming during Parental-to-Zygotic Transition. *Mol Cell* **72**: 673-686.e676.

Zhu W, Xu X, Wang X, Liu J. 2019. Reprogramming histone modification patterns to coordinate gene expression in early zebrafish embryos. *BMC Genomics* **20**: 248.

**Zygotic activation of transposable elements during zebrafish early
embryogenesis**

Bo Li, Ting Li, Dingjie Wang, Ying Yang, Puwen Tan, Yunhao Wang,
Yun-Gui Yang, Shunji Jia, Kin Fai Au

Responses to reviewers

Reviewer #1 (Remarks to the Author):

The authors have adequately answered all my points.

Reviewer #2 (Remarks to the Author):

The authors went above and beyond in their revisions to this paper. All my concerns were satisfactorily addressed and I have no further comments.

Reviewer #2 (Remarks on code availability):

The code is complete and easy to read. I have no further issues with it.

Reviewer #3 (Remarks to the Author):

In revision, the authors put a lot of effort into adding additional experiments and analyses to support their claims. Unfortunately, I do not think these efforts have been adequate relative to how the paper is presented. Although this study still provides the field with potentially useful raw data in the long-read sequencing and characterization of TE expression dynamics, their major claims of biological significance are unwarranted at this time: namely, (1) the data are still insufficient to support the existence of a distinct event of TE activation separate from ZGA, and (2) the phenotypes associated with the mutants cannot as yet be attributed to TEs. Especially regarding the second point, the manuscript seems to be written in a way that downplays and/or neglects important details that are necessary for a full, objective interpretation of the results, which is certainly not ideal for the reader.

Response summary

We greatly appreciate the reviewers' appreciation of our efforts in the previous revision and also we sincerely apologize for the unclear descriptions in the manuscript that may have led to misunderstandings. In this revision, we carefully follow the reviewer's kind suggestions to revise the result presentations aiming to clarify the conclusion in a rigorous manner, including:

- (1) Only describing the omics features of the TE-alone transcript activation and removing the conclusion of ZTA as a "separate" event from zygotic genome activation.
- (2) Clarifying that the mutants' phenotype is not associated with TE insertions, although we presented these data aiming to improve the completeness of transcriptomic profile over zebrafish early embryogenesis.

For your convenience, the edits are highlighted in red. Please feel free to let us know if more edits are needed and we will make the best efforts to accommodate them.

Main issues:

- The interpretation of the mutants and their phenotypes remains problematic relative to the potential roles of TE-gene fusions. The TE in "zeat1" is indeed in the UTR (which the authors still do not make clear in either the main text or Fig 6a; they bury this fact in the supplement), so the phenotype of the LOF mutation that disrupted the ORF would by default be attributed to loss of the protein unless demonstrated otherwise. "zeat2" is indeed annotated by RefSeq as fastkd5, and the ORF is not interrupted by any TE either (in fact, there is no TE annotated in zeat2 in Fig 6A, nor on the UCSC Genome Browser repeatmasker track when navigating to the provided coordinates) -- why is zeat2 even a gene of focus for the authors?

Response: We sincerely apologize that our revised manuscript still caused misunderstandings and confusions. Please see the following point-by-point clarifications below:

Issues related to *zeat1*.

In the previous revision, we added "*A Tc1N1_DR2 transposon is inserted at the last exon of zeat1 without interrupting the coding sequence...*" aiming to convey clearly that the TE insertion does not affect protein function. We are sorry that we did not replace the term "the last exon" with "3' UTR". Thank you for the kind reminder. In

this revision, we follow the reviewer's suggestion to revise the sentence (Page 24, Line 9-10) to avoid potential misleading:

"A Tc1N1_DR transposon is inserted at the 3' UTR of zeat1 without interrupting the coding sequence..."

Issues related to zeat2.

Our original design of the project was to utilize long-read RNA sequencing to enhance the profiling of transcriptomic dynamics during zebrafish embryonic development --- although many interesting results of TEs led to the focus of manuscript on TEs, we also discovered previously unannotated genes, some of which has critical functions. A postdoc analyzed the data in 2019 and filtered the expressed loci against the widely-used reference annotation library Ensembl to define "novel/unannotated genes", which did not report the loci of *zeat1* and *zeat2*. At that time, we identified a few "unannotated" loci, including *zeat1* and *zeat2*, as the hub genes in the co-expression network analysis. Later, the other postdocs took over these targets and found important phenotype of *zeat2* but we forgot to revisit and double check the locus annotation with the other reference library, such as RefSeq, when we prepared the manuscript in 2023. We apologize for the mistake sincerely.

As the critical phenotype of *zeat2* is not yet reported in zebrafish, we think it will be still of value to timely report these results to the community but in this revision we clarify the annotation status of *zeat2* as kindly suggested by the reviewers (Page 25, Line 12-14). However, if the reviewer strongly feels that the results of *zeat2* should be omitted, we are open to removing them.

We are grateful for the reviewer's kind reminder that the previous presentation of the results could be misleading regarding the non-existent association between TE insertion and observed phenotype of mutants. To avoid the misleading, we separate the characterization results of TE-gene transcripts and functional study of *zeat1* and *zeat2* into two sections "Characterization of TE-gene transcripts" (Page 23, Line 3-14) and "Novel loci identified with essential functions for zebrafish early embryogenesis" in the revised manuscript and add details of how we selected *zeat1* and *zeat2* for study (Page 23, Line 15- Page, 24, Line 1-2).

- The justification for a distinct ZTA from ZGA is still not adequate. Fig 2A is potentially misleading in this regard (see below) and at odds with Fig 2D, which seems to show the TE genes and non TE genes pretty much span the same range of

expression levels. To demonstrate a distinct ZGA, the authors would need to graph **distributions** of expression levels in each gene group at each time point and show statistically significant differences between groups. Regarding Fig 2A -- the top graph appears to show there are overall far fewer TE-alone transcripts than the other categories (and in fact, the overlapping y-axis scales are very misleading in this regard). The bottom graph appears to show summed expression levels across all genes in a group, which have very different numbers of genes. So, the sum will scale with the number of genes included in a group, thus the largest groups will have the highest sum. A bulk-level analysis like this that ignores group sizes and variation within groups is not adequate to support the authors' claims of a distinct ZTA. Related, Page 9 Line 1 -- citation 24 is a review paper and thus is not the primary source for the timing of minor/major ZGA. At 64 cell, only a single (very unique) locus is activated (mir-430), and minor wave proper is considered to be 128c-512c. (Heyn Cell Rep 2014, Hadzhiev Dev Cell 2023). In concordance with this, a more recent characterization (Bhat Cell Rep 2023) did not identify robust non mir-430 nuclear transcription at 64c_(they suggest spurious/stochastic expression occurs prior to this)

Response: Thank you for the comments and suggestions. The response and revision are summarized into three parts:

1. To find more evidence of the difference between ZTA and ZGA:

We are grateful for the reviewer's suggestion to find more solid evidence to examine the difference of expressions of TE-alone transcripts vs. others with statistical significance. Also, we agree that "A *bulk-level analysis like this that ignores group sizes...*". Therefore, here we follow the reviewer's thoughtful suggestion "to graph **distributions** of expression levels in each gene group at each time point and show statistically significant differences between groups." (please see **Response Figure 1** below, which replaces the bottom panel of Fig 2A in the revision): we generate a boxplot of expression distribution of each group (i.e., TE-alone, TE-gene and gene) at each stage and performed Wilcoxon rank sum tests to compare expression difference between groups. This suggested analysis shows that TE-alone transcripts have significantly different abundance distribution compared to TE-gene and gene groups across all tested stages, except for the 64-cell stage (Response Figure 1). In

details, this suggested analysis provides two key observations to distinguish ZTA from ZGA

- Before the oblong stage, TE-alone transcripts have significantly lower expression levels than TE-gene and regular gene transcripts.
- After the oblong stage, TE-alone transcripts have dramatically increased expression levels, greater than those of TE-gene and gene transcripts.

Thanks to the reviewer's suggestion, we add these results at Page 9, Line 2-7 in the revised manuscript.

2. Clarifications regarding Fig. 2a and Fig. 2d:

While Fig. 2a was generated using our long-read RNA-seq, Fig. 2d was generated by a public short-read RNA-seq data of zebrafish embryo samples including WT, α -Amanitin injection, cycloheximide treatment, triple LOF and mRNA rescue samples (PRJNA206070). The differences between two sequencing techniques may lead to different estimates of absolute abundance and thus cause more or less difference in the details of expression patterns. Although the extent of these differences is yet evaluated in a comprehensive way by the community, both long-read RNA-seq and short-read RNA-seq have been adopted widely without significant scientific conflict. In fact, the increased number of red dots (i.e., the number of active TE-alone transcripts) and the abundance represented by red dots (i.e., the expression levels of active TE-alone transcripts) in Fig. 2d from the sphere to the shield stages is consistent with the pattern in Fig. 2a.

3. Regarding the minor wave of ZGA:

We appreciate the reviewer's insightful comment and agree that it is important to cite original studies regarding the timing of minor ZGA. To address this, we have revised the manuscript to include relevant primary literature and clarified our interpretation.

We recognize that the precise timing of minor ZGA remains a topic of debate, largely due to the elegant findings showing that, at the 64-cell stage, transcription is predominantly limited to the unique locus mir-430 (Hadzhiev et al., 2023; Bhat Cell Rep 2023). We previously referred to the 64-cell stage as the earliest recorded ZGA, aiming to set up a reference time point for comparison with ZTA timing. In fact, even if the minor wave initiates between the 128- and 512-cell stages, our data still convincingly show that TE-alone transcripts are activated later than regular genes. To enhance rigor in the revised manuscript, we have cited the relevant publications and clarified in the revised manuscript that the minor wave of ZGA may begin between the 64-cell and 512-cell stages (Page 9, Line 1-2).

Response Figure 1. Comparison of transcript abundance among gene, TE-gene and TE-alone groups. Wilcoxon rank sum test is applied. ****, 0.0001; ***, 0.001; **, 0.01; *, 0.05. Not significant testing results are not shown in the figure.

Additionally, Page 28 Line 11 - the caveat regarding cytoplasmic polyadenylation needs to be stated much earlier, as a large proportion (all?) of the early increases the authors observe (Fig 2A) are likely due to poly(A) tail lengthening of maternal transcripts rather than de novo transcription. This strongly impacts the interpretation of the dynamics observed. In their response to Reviewer 2, the authors do an analysis with a parallel poly(A)+/rRNA deplete RNA-seq dataset to show that TE transcripts are not influenced by poly(Ad) dynamics, though the issue remains that many other mRNA are affected, and this has an impact on comparisons between the

different gene classes (Fig 2). PS -- other parallel poly(A)+/rRNA deplete datasets exist that *do* have replicates, e.g. Vejnar et al Genome Res 2019.

Response: We are grateful that both Reviewer 2 and Reviewer 3 appreciated our analysis in the previous response showed “*TE transcripts are not influenced by poly(Ad) dynamics..*”. With the Reviewer 3’s kindly suggested data with replicates, we here are able to generate more rigorous results (see details below and **Response Figure 2**) to evaluate the impact on TE transcripts as well as other mRNAs that the reviewer has the concern with:

Using the DESeq2 pipeline, we identified differentially expressed genes between two library preparation methods: poly(A) enrichment and rRNA depletion (Response Figure 2). The result of TE transcripts is similar with our analysis with the single-replicate dataset in the previous revision. From the 1-cell to the 75%-epiboly stages, the number of regular genes potentially affected by cytoplasmic polyadenylation (i.e., genes with higher expression levels in rRNA depletion data in Response Figure 2) is only of a small portion of the total expressed genes, and decreases dramatically as embryo development proceeds. Therefore, the overall results of regular genes and the comparison results with TE transcripts are not dramatically affected by these small portion of targets.

Of note, our analyses, including those involving regular genes, have a step to explicitly remove maternal transcripts and only consider newly zygotically activated transcripts, thereby minimizing the potential impact of cytoplasmic polyadenylation. Therefore, our results are not significantly impacted by the poly(A) selection sequencing approach.

Response Figure 2 is now added to the revised manuscript as the Supplementary Fig. 21; the analysis details can be found at the subsection “25. The comparison of RNA-seq data generated by different library preparations.” of the section “Method” (Page 48, Line 17-19); and the corresponding discussion can be found at Page 29, Line 10-18.

Response Figure 2. Differential analysis between poly(A) enrichment and rRNA depletion library preparation methods. Gene-sig: significantly differential expressed genes are highlighted in red color; Gene-non: insignificantly differential expressed genes are highlighted in grey color; TE-sig: significantly differential expressed TE-alone are highlighted in green color; and TE-non: insignificantly differential expressed genes are highlighted in black color. Down: down-regulated TE and genes in poly(A) enrichment vs. rRNA depletion, which are potentially caused by cytoplasmic polyadenylation; Up: down-regulated TE and genes in poly(A) enrichment vs. rRNA depletion and Non-sig: TE and genes with no significant changes. The counts are listed for each stage (TE & gene).

- If I'm understanding things correctly, there is something off about the N/C proportion analysis that needs to be clarified. The N/C proportion equation (Fig 4A) would seem to suggest that a strongly cytoplasmic TE would have a very small numerator and a larger denominator, leading to a proportion $\ll 1$. Fig 4B is a bulk-level plot, though a distributional plot (boxplot, violin plot) would be more appropriate to demonstrate the variability within/between groups, as well as the range of values -- we would expect both strongly cytoplasmic and strongly nuclear genes, and the shape of the

distributions should reflect this. In Fig 4D, the y-axis scale suggests that all the proportions are above 1 ($\log_2 > 0$) -- is this indeed the case? If a smoothing factor was being applied (e.g., +1?), that would heavily change the shape of the distribution and warrant a reanalysis of statistical significance. I would suggest that the N/C proportion formulation needs to be calibrated if the values overall are skewing > 1 so that we better understand what it is representing.

Response: Thank you for these constructive suggestion of replacing the simple bar plot with the distributional plot. We include the corresponding boxplot as the new Fig 4b in the revised manuscript. For your convenience, both the old simple bar plot and the new boxplot are showed below as **Response Figure 3** for comparison. Overall, we can see similar differences among three groups (TE-alone, TE-gene and gene) and thus the results still hold. Indeed, the N/C values of genes are distributed around one, which reflects the reviewer's thoughtful comment that "we would expect both strongly cytoplasmic and strongly nuclear genes".

In Fig 4d, we apologize that the boxplots do not clearly show all extreme values. In fact, a DNA transposon subfamily (Harbinger-6_DR) has $\log_2(\text{N/C proportion}) = -0.2625$ at the dome stage, that is NC proportion = 0.8336. This single outlier does not affect the overall conclusion drawn from Fig 4d. In fact, previous studies reported that certain types of TEs had nuclear enrichment. For example, SINEs and LINEs were found to be enriched in nuclear regions, especially the nuclear lamina, using APEX-Seq technology¹. Similarly, in mouse embryonic stem cells, LINE1 transcripts were predominantly localized in the nucleus². Additionally, TE insertions have been reported to promote the nuclear enrichment of certain long noncoding RNAs in humans³.

When calculating the relative N/C proportions for TE subfamilies, we did not add a smoothing factor (+1) for the \log_2 transformation --- we do not meet the problem of log transformation with zero because the total sum from a group is not zero.

Response Figure 3. Relative N/C proportion for individual genes, TE-Gene and TE-alone transcript. **(A)** The simple bar plot in the previous version of the manuscript; **(B)** The boxplot in the current resubmission (Fig. 4b). The relative N/C proportion for TE-alone transcripts are calculated on subfamily level and those for TE-gene and gene transcripts are calculated on transcript level. The red dash line indicates the relative N/C proportion as 1. Of note, to guarantee the clarity of the plot, the outliers and data points with the relative N/C proportion >20 are not shown.

References

1. Fazal, F.M. *et al.* Atlas of Subcellular RNA Localization Revealed by APEX-Seq. *Cell* **178**, 473-490 e26 (2019).
2. Percharde, M. *et al.* A LINE1-Nucleolin Partnership Regulates Early Development and ESC Identity. *Cell* **174**, 391-405 e19 (2018).
3. Carlevaro-Fita, J. *et al.* Ancient exapted transposable elements promote nuclear enrichment of human long noncoding RNAs. *Genome Res* **29**, 208-222 (2019).